

# Five years of Sentinel-5p TROPOMI operational ozone profiling and geophysical validation using ozonesonde and lidar ground-based networks

Arno Keppens[1], Serena Di Pede[2], Daan Hubert[1], Jean-Christopher Lambert[1], Pepijn Veefkind[2], Maarten Sneep[2], Johan De Haan[2], Mark ter Linden[2], Thierry Leblanc[3], Steven Compernolle[1], Tijl Verhoelst[1], José Granville[1], Oindrila Nath[1], Ann Mari Fjæraa[4], Ian Boyd[5], Sander Niemeijer[6], Roeland Van Malderen[7], Herman G. J. Smit[8], Valentin Duflot[9], Sophie Godin-Beekmann[10], Bryan J. Johnson[11], Wolfgang Steinbrecht[12], David W. Tarasick[13], Debra E. Kollonige[14], Ryan M. Stauffer[14], Anne M. Thompson[14], Angelika Dehn[15], and Claus Zehner[15]

[1]Royal Belgian Institute for Space Aeronomy (BIRA-IASB), Uccle, Belgium
[2]Koninklijk Nederlands Meteorologisch Instituut (KNMI), De Bilt, The Netherlands
[3]Jet Propulsion Laboratory (JPL) - Table Mountain Facility, Wrightwood, CA, USA
[4]The Climate and Environmental Research Institute (NILU), Kjeller, Norway
[5]Bryan Scientific Consulting LLC, Charlottesville, VA, USA
[6]Science and Technology B.V. (s&t), Delft, The Netherlands
[7]Royal Meteorological Institute of Belgium (RMIB) & Solar-Terrestrial Centre of Excellence (STCE), Uccle, Belgium
[8]Forschungszentrum Jülich, Institute of Energy and Climate Research, IEK-8: Troposphere, Jülich, 52425, Germany
[9]Laboratoire de l'Atmosphère et des Cyclones (LACy), Université de la Réunion, Saint-Denis, France
[10]Laboratoire Atmosphères, Milieux, Observations Spatiales (LATMOS/IPSL), Paris, France
[11]Global Monitoring Laboratory, NOAA Earth System Research Laboratory, Boulder, CO, USA
[12]Deutscher Wetterdienst (DWD), Hohenpeissenberg, Germany
[13]Environment and Climate Change Canada (ECCC), Downsview, ON, Canada
[14]Atmospheric Chemistry and Dynamics Laboratory, NASA/GSFC, Greenbelt, MD, USA
[15]European Space Agency/Centre for Earth Observation (ESA/ESRIN), Frascati, Italy

**Correspondence:** arno.keppens@aeronomie.be

**Abstract.** The Sentinel-5 Precursor (S5P) satellite operated by the European Space Agency (ESA) carries the TROPOspheric Monitoring Instrument (TROPOMI) on a Sun-synchronous low-Earth orbit since October 13, 2017. The S5P mission has acquired more than five years of TROPOMI nadir ozone profile data retrieved from the Level-0-to-1B processor version 2.0 and the Level-1B-to-2 Optimal Estimation based processor version 2.4.0. The latter is described in detail in this work, followed

by the geophysical validation of the resulting ozone profiles for the period May 2018 to April 2023. Comparison of TROPOMI ozone profile data to co-located ozonesonde and lidar measurements used as references, concludes to a median agreement better than 5 to 10 % in the troposphere. The bias goes up to -15 % in the upper stratosphere (35-45 km) where it can exhibit vertical oscillations. The comparisons show a dispersion of about 30 % in the troposphere and 10 to 20 % in the upper troposphere to lower stratosphere (UTLS) and in the middle stratosphere, which is close to mission requirements. Chi-square

tests of the observed differences confirm on average the validity of the ex-ante (prognostic) satellite and ground-based data uncertainty estimates in the middle stratosphere, above about 20 km. Around the tropopause and below, the mean chi-square



value increases up to about four, meaning that the ex-ante TROPOMI uncertainty is underestimated. The information content of the ozone profile retrieval is characterised by about five to six vertical sub-columns of independent information and a vertical sensitivity nearly equal to unity at altitudes from about 20 to 50 km, decreasing rapidly at altitudes above and below. The

barycentre of the retrieved information is usually close to the nominal retrieval altitude in the 20-50 km altitude range, with positive and negative offsets of up to 10 km below and above this range, respectively. The effective vertical resolution of the profile retrieval usually ranges within 10-15 km, with a minimum close to 7 km in the middle stratosphere. Increased sensitivity and higher effective vertical resolution are observed at higher solar zenith angle (SZA, above about $60°$), as can be expected, and correlate with higher retrieved ozone concentrations. The vertical sensitivity of the TROPOMI tropospheric ozone amount

is found to depend on solar zenith angle, which translates into a seasonal and meridian dependence of its bias with respect to reference measurements. A similar although smaller effect can be seen for the viewing zenith angle (VZA). Additionally, the bias is negatively correlated with the surface albedo for the 6-12 km ozone subcolumn, despite the latter's apparently slightly positive correlation with the retrieval degrees of freedom in the signal. For the five years of TROPOMI ozone profile data that are available now, an overall positive drift is detected for the lowest three subcolumns (0-18 km), while a negative drift is

observed for the two subcolumns above (18-32 km), resulting in a negligible vertically integrated drift.

## 1   Introduction

Atmospheric ozone is an Essential Climate Variable (ECV) monitored in the framework of the Global Climate Observing System (GCOS). This is motivated by stratospheric ozone's decisive impact on the radiation budget of the Earth, while tro-

pospheric ozone is the third most important anthropogenic contributor to greenhouse radiative forcing. Ozone moreover plays a central role in the oxidation chemistry of the atmosphere (Monks et al., 2015), and is an important air pollutant. Exposure to high levels of ozone can cause respiratory issues and be detrimental to health, vegetation and materials. Studies related to atmospheric composition and the Earth's radiation budget therefore require accurate monitoring of the horizontal and vertical distribution of ozone at the scale of interest (WMO, 2010).

Atmospheric ozone concentration profile data records have been retrieved from back-scattered solar ultraviolet radiation measurements by nadir viewing satellite spectrometers since the 1960s, starting with the USSR Kosmos missions in 1964-1965 (Iozenas et al., 1969) and NASA's Orbiting Geophysical Observatory in 1967-1969 (Anderson et al., 1969) and Back Scatter Ultraviolet (BUV) on Nimbus 4 in 1970-1975 (Heath et al., 1973), and continuing with the Solar BUV series (SBUV(/2)) after 1978 (Heath et al., 1975). Global Ozone Monitoring Experiment (GOME) in 1995-2002 (Burrows et al., 1999) paved the way

to new generation sounders including SCanning Imaging Absorption spectroMeter for Atmospheric CHartographY (SCIA-MACHY) on board of ENVISAT (Bovensmann et al., 1999), Ozone Monitoring Instrument (OMI) on the EOS-Aura satellite launched in 2004, the GOME-2 instruments on the Meteorological Operational (MetOp) satellites since 2006 (Munro et al.,





2016; Hassinen et al., 2016), and the Ozone Mapping Profiler Suite (OMPS) nadir series started in 2011 (Flynn et al., 2006). Ensuring the uninterrupted continuation of this global monitoring of ozone and other trace gas concentrations is a requirement
of the European Earth Observation Programme Copernicus (Ingmann et al., 2012): the Copernicus Space Component plans a series of three Sentinel-5 atmospheric composition missions, jointly developed by the European Space Agency (ESA) and the European Commission, and to be operated by the EUMETSAT aboard its satellites MetOp-SG-A1 (Meteorological Operational - Second Generation), -A2 and -A3, scheduled in 2024, 2031 and 2038, respectively. As a gap filler between the heritage satellites and the upcoming Sentinel-5 series, Sentinel-5 Precursor (S5P) was launched in October 2017, with TROPOspheric
Monitoring Instrument as unique payload (TROPOMI, Veefkind et al., 2012).

The first atmospheric composition mission of the European Union's Copernicus Earth Observation Programme, the Sentinel-5 Precursor satellite is dedicated to the global atmospheric monitoring and study of air quality, climate forcing, ozone, UV radiation, and volcanic hazards. On board of the S5P early afternoon polar-orbiting satellite, the imaging spectrometer TROPOMI performs nadir measurements of the Earth's radiance from the UV-visible to the short-wave-infrared spectral ranges
at a much finer spatial resolution than its predecessors do, and from which the global distribution of several atmospheric trace gases is retrieved daily, including stratospheric and tropospheric ozone. Developed at the Royal Netherlands Meteorological Institute (KNMI) and based on the optimal estimation method, TROPOMI's operational ozone profile retrieval algorithm has been implemented into the S5P Payload Data Ground Segment (PDGS), providing both near real-time (NRTI, providing 50 % of the pixels by following a checkerboard pattern) and offline (OFFL) Level-2 ozone profile data that is freely accessible from
the Copernicus Data Space Environment Open Hub (https://dataspace.copernicus.eu/).

The first objective of this article is to provide an extensive description of the ozone profile retrieval processor currently used for S5P operational data processing (Section 2). This processor has also been used to back-process the entire TROPOMI Level-1B version 2 data record, providing a homogeneous five-year record of Level-2 ozone profile data, denoted as L2__O3__PR version 2.4.0. The second objective of this work is to summarize the comprehensive quality assessment (QA) of this TROPOMI
ozone profile data record, whereby results are collected from both the ESA/Copernicus Atmospheric Mission Performance Cluster's operational Validation Data Analysis Facility (ATM-MPC VDAF) and the S5P AO Validation Team (S5PVT) activity CHEOPS-5p. The QA approaches used are detailed in Section 3. They are based on the same validation practices as developed for and applied to the heritage sounders (Keppens et al., 2015, 2018). The validation methodology relies on the analysis of data retrieval diagnostics and on comparisons of TROPOMI data with co-located ground-based measurements used as a reference.
The latter are acquired by ozonesondes contributing to WMO's Global Atmosphere Watch (GAW) and its contributing networks NDACC and SHADOZ (Thompson et al., 2019), by tropospheric lidars affiliated with the Tropospheric Ozone Lidar Network (TOLNet, Leblanc et al., 2018), and by stratospheric lidars affiliated with the Network for the Detection of Atmospheric Composition Change (NDACC, De Mazière et al., 2018). QA results are collected in Section 4. The dependence of TROPOMI's ozone profile information content and uncertainty on several influence quantities (like solar zenith angle and surface albedo)
and measurement parameters (like the scan angle) is examined and discussed. This work concludes with an examination of the product compliance with mission requirements and consistency with other TROPOMI ozone retrievals, enabling data users to



verify the fitness-for-purpose of the operational S5P ozone profile data. Conclusions are provided in Section 5, followed by the Appendix that collects some more detailed tables and figures.

## 2 Operational TROPOMI ozone profiling

### 2.1 Instrument description and status

The TROPOspheric Monitoring Instrument (TROPOMI, Veefkind et al., 2012) is the unique payload on the Copernicus S5P satellite, the first atmospheric composition mission in the European Union Copernicus Earth Observation Programme (Ingmann et al., 2012). S5P was launched into a Sun-synchronous low-Earth orbit on Friday 13 October 2017 and, with a foreseen mission lifetime of seven years, is still fully operational today. The ascending node of the satellite orbit crosses the Equator at 1:30 PM local solar time. The four imaging spectrometers of TROPOMI measure the spectral radiance scattered at nadir from the sunlit part of the atmosphere and the solar spectral irradiance, in the 270–2385 nm wavelength range at 0.2–0.5 nm resolution. The field of view at nadir produces ground pixels of $5.5 \times 3.5 \, \text{km}^2$ (along × across track) since the pixel size switch of 6 August 2019 and of $7 \times 3.5 \, \text{km}^2$ before. The large swath width of 2600 km across track produces a nearly daily coverage of the global (sunlit) atmosphere, with narrow gaps between orbits at the Equator. After spectral and radiometric calibration of the Earth radiance and solar irradiance data (Kleipool et al., 2018; Ludewig et al., 2020), operational data processors retrieve the total column abundance of several atmospheric trace gases related to air quality, climate, UV radiation, and environmental hazards. For ozone, a full vertical profile is retrieved.

### 2.2 Ozone profile retrieval algorithm

The operational TROPOMI ozone profile algorithm developed at KNMI derives the ozone concentration as a number density at 33 pressure levels throughout the atmosphere from the TROPOMI reflectance observations in the wavelength region between 270 and 330 nm, provided by Bands 1 (267-300 nm) and 2 (300-332 nm) of the UV detector. In addition to the a-priori ozone profile, the retrieved ozone profiles and their errors, the following diagnostic information is provided: diagonal elements of the a-priori error covariance matrix, a correlation length for the a-priori errors, the a-posteriori error covariance matrix, and the averaging kernel matrix (for the elements corresponding to the ozone profile). The ozone profile is also reported as six sub-columns with a vertical sampling of 6 km up to an altitude of 24 km and lower sampling above until the top of atmosphere (TOA). The main elements of the retrieval algorithm are the forward model (Section 2.2.2), including the state vector and its derivative with respect to the fitted parameters, and the Optimal Estimation (OE) fitting (Rodgers, 2000), described in Section 2.2.4. Before the OE algorithm, several pre-processing steps are applied to the measured spectra (Section 2.2.1). The core of the algorithm is the OE method, which combines the information from the measured spectra with the a-priori information and with the simulated reflectances computed with the forward model calculations. A schematic overview of the algorithm is provided in Figure 1. In this section, a brief overview of the main elements is provided, for a comprehensive description the reader is referred to the Algorithm Theoretical Basis Document (Veefkind et al., 2021).



### 2.2.1 Pre-processing

The pre-processing includes all the steps required to provide recalibrated radiance and irradiance spectra, corrected for po-
larisation and rotational Raman scattering (RRS), that are the inputs for the OE algorithm. It consists of spectral calibration,
spectral and spatial regridding, correction for polarization and RRS, and radiometric correction.

**Spectral calibration**  The spectral calibration is performed only on the solar irradiance data, using the precise knowledge
of the Fraunhofer lines in the solar spectrum (van Geffen et al., 2022). A wavelength shift parameter is fitted on the
irradiance spectrum for the spectral window (270-320 nm), while the assigned wavelengths from the L1B processor are
used as-is for the Earth radiance spectra.

**Spatial regridding**  The ozone profile algorithm uses data from TROPOMI Bands 1 and 2, both registered by the same UV
detector (Kleipool et al., 2018; Ludewig et al., 2020). To suppress noise at shorter wavelengths, the Band 1 data are
measured at a lower spatial sampling in the across-track direction. For the ozone profile retrieval algorithm, the Band 1
and Band 2 data need to be spatially co-registered. Therefore, the radiance and irradiance data for the Band 2 ground
pixels are averaged to match the Band 1 ground pixels. We compute the signal-to-noise ratio (SNR) of the averaged
Band 2 pixels assuming that it is dominated by shot noise. Similar to the radiance and irradiance data, we also average
the Band 2 instrument spectral response function (ISRF) using the same procedure. After the regridding in the across-
track direction, we average the radiances over five scan lines in the flight direction. After the spatial regridding, we have
continuous radiance spectra in the fit window (270-330) nm, for 77 across-track ground pixels with spatial resolution of
$28 \times 28$ km$^2$ in nadir after August 6, 2019 and $28 \times 35$ km$^2$ before that date.

**Spectral regridding**  The spectra of TROPOMI Band 1 and 2 have a very large spectral oversampling (ratio of spectral reso-
lution and spectral sampling) of more than 6.9. The algorithm performs line-by-line radiative transfer computations on
the spectral grid of the measured spectra. To reduce the number of line-by-line calculations, three spectral pixels are
averaged, resulting in an oversampling ratio of at least 2.3. Similar to the averaging in the spatial directions, the SNR is
re-computed assuming shot-noise and the ISRF is averaged to take the effect of the spectral averaging into account.

**Signal-to-Noise ratio**  The final SNR is clipped to a maximum of 150, thus implementing a noise floor. This noise floor has
been introduced to account for errors in the forward model that are larger than the noise and significantly improves the
convergence of the algorithm.

**Polarization and Raman correction**  To be able to run the on-line radiative transfer in scalar mode and only accounting for
elastic scattering, we apply a correction for polarization and RRS. The correction is based on a large dataset for which
we computed the spectra with and without these two effects. A neural network has been trained on this dataset to predict
the correction factors as a function of wavelength, sun-satellite geometry, surface albedo and pressure, and total ozone
column.


**Radiometric correction** From comparisons to other satellite sensors as well as to forward models, it is known that the on-ground calibration performed for TROPOMI Bands 1 and 2 has a significant spectral and viewing angle dependent bias. Also, the instrument shows significant optical degradation in the UV, which is not fully corrected in the Level 1b data. For this reason, a yearly correction (known as soft-calibration) has been implemented which is based on a comparison of the measurements with forward model results. The correction parameters are also updated yearly to follow the instrument changes over time due to its optical degradation. They are obtained using a combination of several observations covering different seasons during the year (always with orbits over the Pacific Ocean), and they are computed as a function of the wavelength, across-track pixel number and radiance level. Pressure, temperature, and ozone profiles from the Copernicus Atmospheric Monitoring Service (CAMS) global analysis are used as inputs. Additionally, the CAMS ozone profiles are scaled to match the total ozone column derived from the OMPS total column data (Jaross, 2017). First, the surface albedo is fit in a small spectral window (328-330) nm. Next, the fitted surface albedo is applied to the entire wavelength fit window (270-330) nm for the forward model calculations. Figure 2 left shows the correction implemented per each year (red points) since the beginning of the TROPOMI mission in 2018 (orbit 4165) up to December 2022 (orbit 24482), for 4 ground pixels (15, 25, 45 and 65), the middle radiance bin and at a specific wavelength = 290 nm. The red points represent the yearly correction computed by combining the black and light gray points of the same year. The right panels in Figure 2 show the radiometric correction computed per each year as a function of the wavelength, for the middle radiance bin and for two ground pixels, 25 (top figure) and 45 (bottom figure). The largest corrections are in the Fraunhofer lines, where the radiance signals are quite low. As can be seen in Table 1, for the period between the reactivation of the RPRO processing and the injection of soft-calibration file v2 (July 25, 2022 to January 15, 2023), the previous version (v1) of the soft-calibration file is used. The difference between the two versions is that for v1 only DDS6 (sixth diagnostic data set) orbits were used to compute the correction parameters (as they were the only available at the moment of computation), while for v2 a combination of DDS6 and OFFL orbits was used. It is worth mentioning that the DDS6 data are identical to the RPRO data for all intents and purposes, so that the difference between the two soft-calibration files is mostly related to the higher statistics for v2.

### 2.2.2 Forward model

The forward model computes the Earth's reflectance for the Sun-satellite geometry of a ground pixel, at the spectral resolution of the observations, using the DISAMAR software (de Haan et al., 2022). As can be seen from the schematics in Figure 1, the output of the forward model is the simulated reflectance that can be compared with the measured reflectance, as well as the derivatives with respect to the fit parameters: the ozone profile at 33 pressure levels, the $SO_2$ column, the surface and cloud albedo (each at three wavelengths). The inputs of the forward model include in addition to the model atmosphere parameters: the solar/viewing zenith and azimuth angles from the L1b data and the ISRF. The model atmosphere contains dry air, $O_3$, $SO_2$, a Lambertian cloud, and it is bounded by a Lambertian surface. It is described by a pressure-temperature profile, the ozone profile, the $SO_2$ profile, the surface albedo, and the cloud albedo, fraction, and pressure. The cloud pressure and fraction are derived from the $O_2$ A-band around 760 nm, and the cloud albedo is fitted during the retrieval of the ozone profile. The ozone



profile is described as a number density at 33 pressure levels in the atmosphere. The atmosphere is assumed to be in hydrostatic equilibrium so that an altitude grid can be computed from the pressure-temperature profile. It is noted that the vertical grid for

the model atmospheres is independent from the grid used for the radiative transfer calculations. The radiative transfer model used is based on successive orders of scattering using eight streams, the ozone absorption cross section is from Malicet et al. (1995), and the $SO_2$ cross section is a composite based on Bogumil et al. (2001) and Hermans et al. (2009).

### 2.2.3 A-priori information

For all state vector elements, the OE method requires an a-priori value and its error estimates. The a-priori information is based

on the climatology of Labow et al. (2015), which provides the ozone profile and standard deviation as a function of latitude and total ozone column. This climatology has been adjusted by replacing the values in the troposphere and above the 0.1 hPa level with the median of the values along the total ozone axis, to make the ozone profile independent of the ozone column for these pressure ranges. Forecast ozone columns from ECMWF are used to compute the a-priori profile from the climatology. The errors on the a-priori are based on the standard deviations provided by the climatology, but values are limited to the range

of 20-50 %, while they are always set to 50 % for pressures larger than 250 hPa (i.e., lower altitudes). Additionally, a 6-km correlation length is used to compute the off-diagonal elements of the a-priori error covariance matrix. The TROPOMI DLER (directionally dependent Lambertian-equivalent reflectivity) (Tilstra, 2022) is used to derive the a-priori surface albedo, while for the a-priori cloud albedo the Fast Retrieval Scheme for Clouds from Oxygen absorption bands (FRESCO) is used. For cloud fractions below 0.2, the cloud albedo variance is set to $10^{-8}$, thus effectively fitting only the surface albedo. For cloud

fractions above 0.2, the opposite is done, using an a-priori surface albedo error of $10^{-8}$ and 1.0 for the cloud albedo error. For the $SO_2$ column, we use an a-priori value of 0.01 DU and an a-priori error of 0.1 DU.

### 2.2.4 Optimal estimation fitting

The OE method is used to retrieve the fit parameter values and their errors, assuming the availability of the a-priori estimates and the a-priori error covariance matrix. The OE method assumes Gaussian distributions for all errors. The cost function $f_c$

that is minimized in OE retrievals is defined as follows (Rodgers, 2000):

$$f_c(\boldsymbol{y}, \boldsymbol{x}, \mathbf{S}_\epsilon, \boldsymbol{x}_a, \mathbf{S}_a) = [\boldsymbol{y} - F(\boldsymbol{x})]^T \mathbf{S}_\epsilon^{-1} [\boldsymbol{y} - F(\boldsymbol{x})] + (\boldsymbol{x} - \boldsymbol{x}_a)^T \mathbf{S}_a^{-1}(\boldsymbol{x} - \boldsymbol{x}_a) \qquad (1)$$

where $\boldsymbol{y}$ is the vector of measured reflectance containing values for the different wavelengths; $F(\boldsymbol{x})$ is the vector of calculated reflectances (the forward model); $\boldsymbol{x}$ is the state vector containing the parameters that are to be fitted; $\mathbf{S}_\epsilon$ is the error covariance matrix of the measurement, which is diagonal when the measurement errors are assumed to be uncorrelated; $\boldsymbol{x}_a$ is the a-priori

state vector; and $\mathbf{S}_a$ is the a-priori error covariance matrix. In Equation (1), the a-priori term is important for the retrieval stability. Updates to the state vector are based on the Gauss-Newton method, while the convergence criterion is based on Rodgers (2000, Sect. 5.6.3). After finishing the iterative loop, the output parameters are collected and the $O_3$ sub-columns are computed by integrating the profile over the sub-column altitude ranges. Also, quality flags are computed from the diagnostic information.



## 2.3    Data selection for the present study

This work reports on the validation of five years of TROPOMI operational ozone profiles retrieval between May 1, 2018 (orbit 2818, actually dated April 30, 2018 in UTC time) and April 30, 2023. These data originate from the chain of processors operated by TROPOMI's MPC, including L1B processor version 2.0 (except for five months end of 2022, see below) followed by the NL-L2 processor version 2.4.0 or 2.5.0, and are obtained from the former Copernicus Open Access Hub (https://scihub.copernicus.eu) with collection number 03 combining offline (OFFL) and reprocessed (RPRO) data streams. Important dates and corresponding changes within this five-year period are listed in Table 1. The near real-time (NRTI) stream differs hardly from the offline stream, but follows a checkerboard pattern in its pixel selection since version 2.5.0 (starting March 15, 2023), allowing more rapid data processing and delivery. Except for minor formatting changes, this reduction in sampling is the only difference between the 2.4.0 and 2.5.0 processors.

The operational TROPOMI orbit data files contain, for each individual ozone profile retrieval, the ozone number density on 33 pressure levels, the integrated tropospheric and total ozone columns, and six integrated sub-columns (0-6, 6-12, 12-18, 18-24, 24-32, and 32-82 km). For the validation activities presented here, we consider the station overpass pixels obtained from the MPC Payload Data Ground Segment (PDGS) in HARP format (v1.15, https://atmospherictoolbox.org/harp). Data users are encouraged to read the Product Read-me File (PRF), Product User Manual (PUM) and Algorithm Theoretical Basis Document (ATBD) of this product, available online (https://sentinel.esa.int/web/sentinel/technical-guides/sentinel-5p/products-algorithms).

In order to avoid misinterpretation of the data quality, we follow the PUM recommendation to use only TROPOMI ozone profile retrievals with a QA-value above 0.5, which includes screening of profiles with SZA > 80°. Another diagnostic information which indicates the quality of the fit is the so-called cost function ($f_c$ in Eq. (1)), which is minimized during the optimal estimation retrieval. Moreover, to filter out retrieved ozone profiles (with number density values $n$) that deviate too strongly from their a-priori profile (with values $n_{ap}$) for all 33 levels $l$ combined, an additional rejection criterion is applied:

$$\sum_{l=1}^{33} |n(l) - n_{ap}(l)| > 10^{13} \text{ molecules/cm}^3 \tag{2}$$

## 3    Quality Assessment: Approaches

The operational quality assessment (QA) of the TROPOMI ozone profile data by the ESA/Copernicus ATM-MPC applies the satellite validation protocol initially developed by Keppens et al. (2015) for the GOME-2, SCIAMACHY and IASI missions. It majorly entails: (1) visual inspections of daily maps of S5P ozone data and associated parameters, (2) assessment of the retrieved information content based on the analysis of the averaging kernels associated with each retrieved pixel, and (3) comparisons to independent reference measurements collected from ground-based monitoring networks. Analysis is performed on the native vertical grid of the profile retrieval (33 levels, see Section 2), although the derived product consisting of six integrated ozone subcolumns is considered here as well.





The daily ozone and correlation checks for a selection of satellite data parameters within the orbit files are produced by KNMI using the PyCAMA software. Such checks provide a view on single-orbit features, correlations between retrievals of subsequent pixels, the appropriateness of the data flagging, etc. The daily results can be found on the TROPOMI MPC Portal for Level-2 Quality Control (https://mpc-l2.tropomi.eu). For the routine comparative validation of the TROPOMI ozone profiles, the automated validation server (VDAF-AVS, https://mpc-vdaf-server.tropomi.eu) deployed within the ATM-MPC VDAF facility automatically collects S5P ozone profiles and correlative measurements to identify suitable co-locations, compare the co-located data, and produce S5P data quality indicators. The VDAF-AVS produces curtain plots (ozone number density as a function of altitude and time) of the satellite data at a selection of ground-based ozonesonde and stratospheric lidar stations, together with curtain plots showing the difference between TROPOMI and the ground-based reference data. The VDAF-AVS also provides statistical estimates of the bias and dispersion of S5P data with respect to the ground-based measurements.

The TROPOMI Validation Data Analysis Facility additionally produces consolidated validation results through the versatile Multi-TASTE validation system developed and operated at BIRA-IASB (Lambert et al., 2014). These consolidated results, including both information content studies and comparisons with independent reference measurements (also see next subsections), are the main focus in this work, and discussed with respect to the TROPOMI product quality targets in the upcoming results sections. Summaries of past and future intermediate results can be found on the VDAF website (https://mpc-vdaf.tropomi.eu) and in the quarterly TROPOMI MPC Routine Operations Consolidated Validation Report (ROCVR) through the same link.

## 3.1 Information content and vertical sounding studies

The information content of the TROPOMI ozone profile data is assessed through algebraic analysis of the averaging kernel (AK) matrix that is associated with each profile retrieval and generated by the same processing algorithm. The row sums of the AK matrix indicate the vertical sensitivity of the ozone profile retrieval. The trace of the AK kernel matrix gives the Degrees of Freedom in the Signal (DFS), to be understood here as the amount of vertical sub-columns with independent information in terms of Shannon information content (Rodgers, 2000). The full width at half maximum (FWHM) of the AK corresponding to a given altitude is selected in this work as an indicator of the effective vertical resolution of the retrieved profile at this altitude (Keppens et al., 2015). This effective resolution of the retrieved information is not the sampling resolution of the vertical grid used for the retrieval process, which usually oversamples the true, physical resolution of the retrieved information. Finally, the effective altitude registration of the retrieved profile information at a given altitude is estimated as the barycentre or peak position of the associated AK at this altitude.

## 3.2 Comparisons with correlative reference measurements

TROPOMI ozone profile data are compared to ground-based reference measurements acquired by instruments contributing to WMO's Global Atmosphere Watch (GAW), the Network for the Detection of Atmospheric Composition Change (NDACC, De Mazière et al., 2018), Southern Hemisphere Additional Ozonesondes (SHADOZ, Thompson et al., 2019), and Tropospheric Ozone Lidar Network (TOLNet, Leblanc et al., 2018): (1) balloon-borne ozonesondes, (2) tropospheric ozone differential



absorption lidars (DIAL), and (3) stratospheric ozone DIAL. The first two reference instruments are used to assess the quality
of TROPOMI profile retrievals in the troposphere and up to the middle stratosphere, while the latter is used as reference
to validate the full stratospheric part of the TROPOMI ozone profile. The ground-based data are collected through ESA's
Validation Data Centre (EVDC, https://evdc.esa.int/) and the respective data host facilities of the ground-based networks. A
global map showing all reference measurement stations considered in this work (55 ozonesondes, 4 tropospheric lidars, 5
stratospheric lidars) is depicted in Figure 3. The geographical distribution of the stations indicates the domain of applicability
of the comparative validation results. The exact station locations and data sources are provided in Tables A1 to A3 in Appendix.

### 3.2.1 Balloon-borne ozonesonde measurements

Launched on board of small meteorological balloons, an electrochemical ozonesonde measures the vertical distribution of
atmospheric ozone partial pressure from the ground up to burst point, typically around 30 km. Measurement errors depend on
the type and the preparation of the sonde instrument (Smit, 2014; Stauffer et al., 2022) as well as on how the post-processing
of the acquired raw data is done. When standard operating procedures are followed, systematic differences with respect to the
reference photometer at the World Calibration Centre for Ozone Sondes (WCCOS) in Jülich are negligible, with an uncertainty
of up to 5 %, and except in the tropical upper troposphere, random uncertainty is less than 3-5 % below 27 km (Smit, Thompson,
and the ASOPOS 2.0 Panel, 2021). The ozonesonde data originate from the NDACC Data Host facility, the SHADOZ archive,
and World Ozone and UV Data Centre (WOUDC). Since their vertical resolution is much higher than that of the TROPOMI
ozone profiles, they can also be used to estimate the vertical smoothing error of the remotely sensed profiles.

### 3.2.2 Differential Absorption Lidar measurements

Ozone differential absorption lidars (DIAL) can measure the vertical profile of ozone number density in the troposphere (500
m a.g.l. to 12-15 km) or in the stratosphere (8-10 km to 45-50 km). Ground-based systems perform network operation in
the framework of the international Network for the Detection of Atmospheric Composition Change (NDACC) and the North
American-based Tropospheric Ozone Lidar Network (TOLNet). For stratospheric measurements, effective vertical resolution
typically degrades with altitude from a few hundred meters at 10 km to 3–5 km in the upper stratosphere, and the total
uncertainty (systematic and random effects included) ranges from 4 % below 30 km to 10 % or more at 35 km and above
(Leblanc et al., 2016). For tropospheric measurements, effective vertical resolution also degrades with altitude, from a few
meters in the boundary layer to 2-3 km in the upper troposphere, and the total uncertainty ranges from 4 % (profile bottom)
to 10-20 % (profile top). Between about 3 and 10 km, tropospheric ozone lidar measurements show a mean difference with
ozonesonde of less than 2 % and a root-mean-square deviation below 3 %, which are well within the combined uncertainties
of the two measurement techniques (Leblanc et al., 2018). The MPC VDAF and its AVS make use of DIAL ozone profile data
available through EVDC, originating from the NDACC Data Host facility, and through the TOLNet data archive.



### 3.2.3 Spatiotemporal co-location of data pairs

Comparisons between retrieved satellite pixels and reference measurements are based on unique spatiotemporal co-locations within 35 km and 12 hours. This means every observation, whether by TROPOMI or a reference instrument, is considered in at most one comparison. The 35 km radius is chosen to have the ground-based reference measurement typically within the overpassing $28 \times 28\,\text{km}^2$ satellite pixel, unless this pixel is flagged bad. In that case the closest unflagged neighbouring pixel is picked. The $\pm 12$-hour time window selects same-day (24 hour period) observations for both daytime and nighttime

measurements. Figure 4 displays the distribution of the temporal mismatch for all collocations considered in this work. About 40 % of the ozonesonde collocations and close to all tropospheric lidar measurements take place within one hour of the satellite overpass. The ozonesonde profile time stamp is taken as the launch time for a few hours flight, which is mostly taking place before local noon (positive satellite minus reference time). The DIAL time stamp is the middle of the minutes to hours measurement integration time window. This results in an eight hours difference on average for the nighttime stratospheric lidar

observations that are mostly taking place before midnight the same day (negative time difference).

### 3.2.4 Data harmonisation and comparison

The comparative validation of the TROPOMI ozone profiles with respect to reference measurements requires calculating difference profiles, and hence the harmonisation of satellite and reference profiles in terms of physical quantities and vertical sampling at least (Keppens et al., 2019). The reference measurements are first converted to the altitude-number density repre-

sentation if needed, using auxiliary data that is provided with the ozone profile data (Keppens et al., 2015). Next, the reference measurements, which are acquired at higher vertical resolution than the TROPOMI profile data, are regridded to the satellite retrieval grid. This is achieved using a mass-conserving regridding approach, meaning that the integrated ozone column amount of each profile is conserved throughout the operation (Langerock et al., 2015).

Given that the effective vertical resolution of the satellite retrieval is significantly lower than the resolution of the retrieval

grid (also see information content studies), the retrieved profiles come with a vertical smoothing error $\epsilon_V = (A - I)(x - x_a)$, with $A$ and $x_a$ the retrieval's averaging kernel and prior profile, respectively. This vertical smoothing error is a systematic error that is correlated with the true state $x$, and will therefore show variations on the same spatiotemporal scale as the actual ozone field. However, assuming the reference observations not to be vertically smoothed at all, the vertical smoothing *difference* error in the profile comparisons is only given by the satellite retrieval's vertical smoothing error $\epsilon_V$. This means it can be accounted

for by smoothing of the reference profiles $x_r$ with TROPOMI's averaging kernels before comparison (Rodgers and Connor, 2003; Keppens et al., 2019):

$$x'_r = Ax_r + (I - A)x_a \tag{3}$$

which is indeed equivalent to correcting the profile's difference for the vertical smoothing (difference) error $\epsilon_V$, if the reference profile $x_r$ is taken as the best and non-smoothed estimate of the true profile $x$: $x'_r = x_r + \epsilon_V = x_r + (A - I)(x - x_a) \equiv Ax_r + $

$(I - A)x_a$ for $x \equiv x_r$ (Rodgers, 2000). After application of Eq. (3) to each reference profile, making use of the collocated





satellite profile's averaging kernel matrix and prior profile, the difference in satellite and reference ozone number density values is calculated for a selection of influence quantities.

Chi-square tests ($\chi^2 = \Delta x^T S_\Delta^{-1} \Delta x$) are added to all difference calculations. They allow verifying whether the observed differences $\Delta x$ between the satellite and reference profiles are consistent with the ex-ante (predicted) uncertainties on the
difference $S_\Delta$ (von Clarmann, 2006; Keppens et al., 2019). The latter contains the satellite and reference covariances, and uncertainties that are due to sampling, smoothing, and retrieval differences. By application of the vertical averaging kernel smoothing, however, retrieval differences and vertical sampling and smoothing differences are essentially removed from the difference covariance (Keppens et al., 2019). This means that for the results presented here the difference covariance mainly contains the ex-ante satellite and reference covariances. Horizontal and temporal sampling and smoothing differences are
moreover minimized by application of the strict collocation criteria (see previous subsection).

## 4  Quality Assessment: Results

### 4.1  Data content studies

The TROPOMI MPC quality control portal creates daily global maps of the six partial columns provided in the ozone profile product, together with the integrated total column. The latter is compared with the daily global map of the TROPOMI total
column retrieval to assess their mutual consistency. Daily global maps easily allow identifying data gaps, retrieval and data screening artefacts, along-orbit striping, and other large-scale features that are not typically detected through comparison with respect to point-like ground-based reference data. The monthly mean of the six ozone profile sub-columns for October 2020 is shown in Figure 5. Each sub-column, starting from the bottom-left, shows the main layer features: for example, in the first sub-column from 0 to 6 km the low ozone levels highlight the Himalaya mountain range in Asia and the Andes mountain range
in South America, while in the fifth sub-column from 24 to 32 km the stratospheric ozone layer is clearly visible. Note that in the absence of clouds, data files sometimes contain negative surface albedo values. The TROPOMI ground pixels affected by this anomaly are usually located at the east and west edges of the across-orbit measurement swath. For now, these negative values are set to zero in the radiative transfer code and validation tools (as an influence quantity), but not in the ozone profile data distribution to users.
A comparison between the a-priori ozone profile and the retrieval is shown in the zonal averages in Figure 6 for two days in 2020, one in Spring when there is the highest amount of ozone, and one during the ozone hole season. Profiles are flagged bad if this difference exceeds the criterion in Eq. (2), which typically occurs towards the edge of the swath, i.e., for high viewing-zenith angles. Figure 7 shows the percentage of retrieved pixels with good quality after filtering out those pixels with QA < 0.5, cost function > 200, and the prior condition expressed by Eq. (2), applied in this specific order. Each line represents the
average of the same orbit number over all the days of the specific month (March, July, October, or December). The orbits are numbered in ascending order from left to right, as shown in the right side of Figure 7. There is a slight seasonal dependence, which also changes over the years. The daily variation shows a distinguished minimum around orbit 9-10, when TROPOMI passes over the South Atlantic Anomaly. This is more visible in the months of March, July and October, while it does not have





a large effect in December. In the Appendix, Figure A1 shows the percentage trend of each separate filter applied in the same
four months. The first two filters (QA and cost function) behave similarly, while the condition in Eq. (2), in the third column,
rules out only a small percentage, as expected.

## 4.2   Vertical sensitivity and independent information

The information content of the TROPOMI ozone profile data is assessed through the analysis of the averaging kernel matrices.
The retrieval's degrees of freedom in the signal (DFS) as a measure of the number of independent pieces of information is given
by the matrix trace. The monthly mean DFS of the ozone profile is shown in Figure 8 for October 2020, during the ozone hole
season. It shows a high correlation with the amount of ozone (and hence with latitude), as can be expected for an OE retrieval:
higher ozone amounts yield higher spectral signals and therefore have a positive impact on the retrieval sensitivity. Overall, the
retrieved information on ozone is distributed over five to six vertical sub-columns of independent information: about 5.5 on
average, with values closer to six in the mid-latitudes and values just above five in the tropics and towards the poles.

The ozone retrieval DFS are also assessed for the six subcolumns provided in the TROPOMI ozone profile product separately.
Figure 9 shows the correlation between each subcolumn DFS and ozone amount (first column). Subsequent plot columns show
the layer-DFS as a function of latitude, time, solar zenith angle (SZA), viewing-zenith angle (VZA), cloud fraction, and
surface albedo, including the subcolumn DFS mean and quantiles. First, it is clear that the six provided ozone subcolumns
do not match the five to six pieces of information in the retrieval. The highest column (32-82 km, top row) roughly has 3
DFS, the column below (24-32 km) 1 DFS, then two columns (12-18 and 18-24 km) each have about 0.5 DFS, and finally the
lowest columns have 0.3 and 0.2 DFS from 6-12 and 0-6 km (bottom row), respectively. Together, these add up to the average
of 5.5 indeed, but significant deviations can be observed. Especially the 12-18 km subcolumn DFS highly correlates with its
ozone amount ($R = 0.8$) and the solar zenith angle, resulting in a strong meridian dependence as well. A similar but much
reduced dependence meredian is seen for the two adjacent subcolumns, while the correlation with ozone, SZA, and latitude
becomes slightly negative for the lowest column, and for the highest column for high solar zenith angles in combination with
high surface albedos. The latter typically occurs for retrievals above Antarctic sea-ice, which, due to the S5P orbit inclination,
usually take place towards the left end of the swath, as can be seen from the viewing-zenith angle dependence.

Yearly drifts are added to the temporal dependence plots in Figure 9. A DFS or retrieval sensitivity degradation is essentially
non-existent for the lowest three subcolumns. The three columns above, however, show a degradation of just below 0.01 DFS
per degree of freedom per year, corresponding to a 5 % DFS decrease over the five year period for the full profile. The vertical
distinction in this degradation is in agreement with the typically stronger degradation towards the shorter UV wavelengths
used primarily for the stratospheric ozone retrieval. The degradation of the longer wavelengths, relevant for the retrieval of
tropospheric ozone, is less pronounced and hardly affects the corresponding DFS.

The vertical sensitivity of the S5P ozone profile data, determined by the averaging kernel matrix row sums, is assessed from
the plots in Figure 11 (fourth graph in each plot). On average, it nearly equals unity at all altitudes from about 20 km up to
50 km, meaning that the information in the retrieved product fully originates from the TROPOMI observation. However, the
sensitivity decreases rapidly at altitudes below 10 km and above 50 km. Around the tropopause between 10 and 20 km (and





around 25 and 50 km for high SZA), an over-sensitivity larger than one can be observed, which is a rather typical compensating effect for the under-sensitivity below (and above) in nadir profile retrievals. This over-sensitivity seems to be more pronounced and vertically expanding for higher solar zenith angles, and again correlates with higher ozone concentrations. As a result, the retrieved ozone below about 24 km will also show seasonal and meridian dependencies in its comparison with reference observations (see next sections). The vertical sensitivity usually drops below 0.5 towards the surface, meaning that the majority of the information in the lowest subcolumn originates from the prior profile, rather than from the spectral satellite measurement.

### 4.3 Vertical sounding accuracy

The vertical sounding accuracy of the S5P ozone profile retrieval is assessed through the analysis of individual averaging kernels. The kernel peak position and width provide measures of the effective vertical retrieval altitude and resolution, respectively. When the retrieval information barycentre of an AK differs from the nominal retrieval altitude (Keppens et al., 2015), an offset different from zero is observed in the fifth graph in each plot of Figure 11. In this work, the effective retrieval resolution is estimated from the kernels' Full Width at Half Maximum (FWHM) and plotted for all collocated kernels in each sixth graph of Figure 11. Note that jumps can occur in both quantities due to the finite retrieval grid.

The altitude registration of the retrieved profile information usually is close to the nominal retrieval altitude in the 20-50 km altitude range, i.e., the offset is nearly zero. Yet the retrieval shows increasingly positive and negative offsets below and above the 20-50 km altitude range, respectively, which can reach 20 km towards the surface. This means the majority of the retrieved surface ozone information (ignoring the prior contribution) comes from the UTLS, in agreement with the occurrence of a sensitivity peak at that altitude. The direction of the offset is always towards higher retrieval sensitivities, i.e., positive for the troposphere and negative for the highest altitudes.

The effective vertical resolution of the TROPOMI ozone profile retrieval usually ranges within 10-15 km, with an optimum close to 7 km in the middle stratosphere (around 30-40 km). Better effective vertical resolutions (reduced kernel FWHM) can be observed for higher solar zenith angles (side-ward atmospheric irradiation) in the troposphere and higher stratosphere, as can be expected for nadir (ozone) profilers. On the other hand, the kernel width becomes ill-defined for the very broad averaging kernels that originate from tropospheric level retrievals at lower solar zenith angles (as for all tropospheric lidar comparisons). With only one retrieval degree of freedom up to about 18 km (lowest three subcolumns in the previous section), the retrieved information is indeed very much vertically smeared, and a large part of the tropospheric ozone information comes from the prior profile. The low-SZA retrieval therefore goes hand in hand with increased vertical smoothing errors, as discussed into detail in Section 4.5 on uncertainty validation.

### 4.4 Comparison results

Figure 11 contains all comparisons between TROPOMI ozone profiles and reference data, colour-coded with SZA, and corresponding statistics. Included are level-specific chi-square tests, which are discussed in the next section, and the information content diagnostics that facilitate the interpretation of the comparison results. The S5P ozone subcolumn comparisons in Figure 10, on the other hand, are only with respect to vertically integrated ozonesonde measurements, again with preceding averaging





kernel smoothing. Compared to ozonesonde and tropospheric lidar data, the S5P RPRO/OFFL data has a mean bias below $\pm$5-10 % in the troposphere, although the reduced sensitivity towards the surface indicates that a significant part of the retrieved information originates from the a-priori assumptions. In the stratosphere up to 35 km, stratospheric lidar data comparisons conclude to a mean bias of $\pm$5-10 % as well. The bias goes up to -15 % above (35-45 km), but with vertical oscillations (positive

and negative). These oscillations of the bias may be due to a typically larger a-priori error in the mid and high stratosphere (above 20 %) in comparison with other retrievals. S5P data comparisons with ozonesonde and stratospheric lidar data show a dispersion of order of 30 % in the troposphere, and 10 to 20 % in the UTLS to upper stratosphere.

A solar zenith angle (or essentially optical path length) dependence of the TROPOMI bias is observed for the lowest subcolumns (Figure 10 fourth column), which also translates into a seasonal and meridian dependence of the bias. Whereas there

is an increase of the DFS and bias for the 6-12 km column with SZA, this correlation seems to be somewhat compensated for in the lowest column by increased atmospheric penetration of the sunlight at low solar zenith angles (0 to about 30°). A similar but reduced effect can be seen for the viewing-zenith angle. Additionally, the bias is clearly negatively correlated with the surface albedo for the 6-12 km subcolumn, despite the latter's apparently slightly positive correlation with the retrieval DFS. As a result, increased biases are found for high-SZA observations above highly reflective scenes, like is the case for Antarctic

(sea) ice. On the other hand, when the deviation from the prior profile becomes too strong, these observations are flagged by the check Eq. (2).

For the five years of TROPOMI RPRO/OFFL ozone profile data that are considered in this work, comparison with the ozonesonde data reveal a positive drift for the lowest three subcolumns (0-18 km), while a negative drift is observed for the two subcolumns above (18-32 km). The 6-12 km subcolumn shows the highest temporal difference change, with a positive

drift close to 8 % over the five-year period, which is also caused by higher positive biases in 2022 and early 2023. Overall, no drift is found for the profile integrated from 0 to 32 km (upper row in Figure 10). More detailed, meridian drift assessments are shown in the right column of Figure A2. These plots show robust linear regression results for the temporal dependence of the TROPOMI bias with respect to ozonesonde measurements (again on the retrieval grid). The horizontal bars indicate two-sigma uncertainties on the drift from a bootstrapping technique with thousand samples (Efron and Tibshirani, 1986). The significant

positive and negative drifts that were observed for the subcolumns on the global scale are confirmed here for the tropics and mid-latitudes, with values up to 2-3 %/year below 20-25 km and minus 1-2 %/year above. No significant tropospheric drifts are detected towards the poles.

## 4.5 Validation of uncertainty estimates

The validation of uncertainty estimates essentially consists in verifying the coherence of the ex-ante (prognostic) retrieval

uncertainty estimates using chi-square tests (Rodgers and Connor, 2003). These are here performed after strict collocation and averaging kernel smoothing, meaning that spatiotemporal sampling and smoothing difference errors are mostly corrected for. The retrieval's vertical smoothing error is a systematic error that is correlated with the true state, and therefore shows variations on the same spatiotemporal scale as the actual ozone field. Although corrected for by the kernel smoothing operation, it is here





discussed in comparison with the total ex-ante uncertainty in order to have a view on the typical (average) magnitude of the
smoothing difference error.

The chi-square plots in Figure 11 (third graph in each plot) demonstrate that on average the observed differences confirm
($\chi^2$ close to one) the combined ex-ante satellite and ground uncertainty estimates in the stratosphere, despite the appearance of
large outliers. However, around the tropopause and below (around 15-20 km and lower), the mean chi-square value increases up
to about four for both ozonesondes and tropospheric lidars, with especially high values for the tropics (low SZA) and Antarctic
(high SZA and surface albedo), see Figure A2. Here, the prognostic (random) satellite uncertainty seems underestimated by
a factor of two, assuming correct reference uncertainties as discussed in Section 3.2. This can also be seen in the difference
plots, as the thin dashed lines representing the dispersion of the difference are further away from the mean difference than the
dotted lines representing combined ex-ante uncertainties. Adding the smoothing difference error to the latter results in the full
thin black lines, pointing at values that are typically 10-30 % higher. The largest smoothing errors (up to 50 %) occur in the
UTLS and towards the surface (where kernel edge effects are also at play).

## 4.6  Compliance with mission requirements

TROPOMI mission requirements have initially been expressed in the Science Requirements Document (SRD, Sentinel-5 Pre-
cursor Team, 2008) and the Geophysical Validation Requirements document (GVR, Sentinel-5 Precursor Team, 2014), and
have later been reproduced in the TROPOMI validation plans and the product-specific ATBD (Veefkind et al., 2021). The SRD
focuses on accuracy requirements for the integrated subcolumns, while the GVR provides requirements on the vertical resolu-
tion and on the systematic and random uncertainties of the ozone profile product specifically. All requirements are summarized
in Table 2, with the compliance of the operational TROPOMI ozone profile product added (Fully, Partially, or Not compliant,
although the latter does not occur).

The operational ozone profiles and derived subcolumns are at least partially compliant with all mission requirements. With
accuracy representing the closeness of the satellite observations to the true value as estimated by the reference measurements,
the corresponding requirements are on average fully met for the lowest three subcolumns. This can be seen from Figure 10,
with the black lines (average differences) being within the grey areas (SRD requirements). Due to vertical bias oscillations,
however, reaching up to about 15 %, the subcolumns above 24 km do not always comply with the accuracy requirement of 3
% above 18 km.

The requirements on the bias are less strict (in terms of systematic uncertainty in the comparisons) for the entire profile in
the GVR. The vertical bias oscillations in the stratosphere reaching up to 15 % are still much below the 30 % limit. On the
other hand, the random uncertainty requirement is only met above the UTLS. Around the UTLS and in the troposphere, the
comparison dispersion reaches 30 % and hence does not comply. The vertical resolution only partially complies with the GVR
too. The vertical retrieval grid is sampled at a resolution of 6 km or higher, but the effective vertical resolution of the profile
equals 10 to 15 km on average, and goes down to 7 km at minimum, thus just not meeting the resolution requirement.





### 4.7 Mutual consistency with other TROPOMI ozone products

A straightforward check that is also performed on a daily basis from the TROPOMI Level-2 quality portal, is to verify whether the integrated ozone profile matches the total ozone column retrieval. The integrated ozone column from the sub-columns of the ozone profile product is compared with the vertical column of the TROPOMI GODFIT Total Ozone product (Garane et al., 2019) in Figure 12. It shows that the relative difference between the two columns in the month of October 2020 (RPRO datasets) typically amounts to about 5 %, meaning that the integrated profile slightly underestimates the total column retrieval, although geographical features seem to be well-captured, as for the six subcolumns in Figure 5. Taking into account possible drifts described above, this makes the operational TROPOMI ozone profile product and its (sub)column derivatives suitable for studies of atmospheric chemistry and dynamics, but not for vertically resolved trend studies.

Several scientific TROPOMI ozone profile retrieval products have been developed as well. First by Zhao et al. (2020), using an optimal estimation retrieval algorithm previously applied to GOME, GOME-2, OMI, and OMPS. Mettig et al. (2021) followed with the TOPAS (Tikhonov regularised Ozone Profile retrievAl with SCIATRAN) algorithm, additionally applied to the Aura Microwave Limb Sounder (MLS) and to OMPS for intercomparison. TOPAS has also been used for joint UV-IR retrievals from TROPOMI and the Cross-track Infrared Sounder on the Suomi National Polar-orbiting Partnership (CrIS/Suomi-NPP), as both are flying in loose formation in the same orbit (about three minutes separation) (Mettig et al., 2022). The same combination of TROPOMI UV and CrIS IR retrieval wavelengths has been exploited by the MUlti-SpEctra, MUlti-SpEcies, MUlti-SEnsors (MUSES) retrieval algorithm, which is a core part of the TRopospheric Ozone and its Precursors from Earth System Sounding (TROPESS) pipeline developed by Malina et al. (2022). An additional TROPOMI ozone profile product is currently under development at the Rutherford Appleton Laboratory (RAL, UK) within ESA's Climate Change Initiative (CCI) on ozone. It is based on the existing RAL UV-Vis processor (v2.14 to v3.01, see Keppens et al., 2018; Pope et al., 2023), which is applied to the operational TROPOMI Level-1b data.

Zhao et al. (2020) selected five stations globally to compare their TROPOMI product, based on the operational L1B v1.0, with ozonesonde profiles from March 2018 to December 2019. Their TROPOMI retrieval agreed with the ozonesondes to within ±5 % from 0.8–30 hPa and within ±15 % below 30 hPa, thus performing comparably to the operational retrieval around the UTLS, while somewhat worse below. Their TROPOMI retrievals showed significant reduction in mean biases over the climatological profiles below 30 hPa, while the retrieved ozone profiles showed worse agreement with ozonesonde than the a-priori profile above 20 hPa, mainly due to not using measurement information below 314 nm.

The TOPAS ozone profile retrieval, already based on L1B v2.0, was validated by comparison with ozonesonde and stratospheric lidar data between June 2018 and October 2019 (Mettig et al., 2021). The validation with lidar measurements showed a bias within ±5 % between 15–45 km, with a standard deviation of 10 %, thus doing slightly better than the operational algorithm. The validation with ozonesondes showed comparable agreement in the lower stratosphere, with deviations of less than ±5 % at all latitudes in the altitude range 18–30 km, and a standard deviation of the mean differences of about 10 %. Below 18 km, on the other hand, the relative mean deviation in the tropics and northern latitudes remained within ±20 %. At southern latitudes, larger differences of up to +40 % occurred between 10 and 15 km. The standard deviation is about 50 % between





7–18 km and about 25 % below 7 km. The combined TROPOMI-CrIS TOPAS retrieval showed reduced mean differences and standard deviations with respect to tropospheric lidar data (limited to the northern subtropical region) in comparison to the UV-only retrieval (Mettig et al., 2022). The validation with ozonesondes showed rather minor improvements.

The MUSES algorithm was used for CrIS-TROPOMI, CrIS-only, and TROPOMI-only ozone profile retrievals from September 2019 to August 2020 (Malina et al., 2022). The TROPOMI-only precision was typically below 5 % in the troposphere,

in agreement with the operational retrieval results. CrIS-only showed a mean bias between 1.4 % and 10.4 %, depending on the season. The performance of the joint retrieval was comparable to that of CrIS-only, with evidence that the joint retrieval provided benefit over CrIS-only with mean biases between 0.2 % and 7.4 %. All three products showed comparable root-mean-square errors on the tropospheric difference of about 20 % or below, which is typically somewhat lower than for the operational TROPOMI product.

**5  Conclusions**

This work reports on the operational retrieval and geophysical validation of Sentinel-5p TROPOMI nadir ozone profile data carried out by the ESA/Copernicus Atmospheric Mission Performance Cluster (ATM-MPC). The NL-L2 processor version 2.4.0/2.5.0 was developed at KNMI and derives ozone number densities at 33 pressure levels from the TROPOMI reflectance observations provided by Bands 1 and 2 of the UV detector, using L1B processor v2. The main elements of the operational

retrieval algorithm are the spectral pre-processing, including soft-calibration, to match both UV bands, the forward model, and the Optimal Estimation fitting. Despite the TROPOMI pixel resolution increase in August 2019 and the soft-calibration changes in July 2022 and January 2023 reducing along-orbit striping, a consistent five-year ozone profile record, ranging from May 2018 to April 2023, is obtained for the entire sunlit Earth.

The comprehensive quality assessment of the TROPOMI ozone profile data record combines results from both the ATM-

MPC Validation Data Analysis Facility and the S5P Validation Team. The prescribed validation methodology includes the analysis of satellite data content, retrieval information diagnostics, and comparisons with co-located ground-based reference measurements. The latter are acquired by ozonesondes contributing to WMO's Global Atmosphere Watch and its contributing networks, by tropospheric lidars from the Tropospheric Ozone Lidar Network, and by NDACC stratospheric lidars. By the application of tight co-location criteria (same-day overpasses) and averaging kernel smoothing of the reference observations,

sampling and smoothing difference errors are reduced to a minimum. Comparison of the TROPOMI ozone profile data with the reference observations then concludes to a median agreement better than 5 to 10 % in the troposphere. The median bias goes up to -15 % in the upper stratosphere, exhibiting vertical oscillations. The comparisons show a dispersion of about 30 % in the troposphere and 10-20 % above. Chi-square tests on these uncertainties demonstrate that the observed differences confirm the satellite (and ground) uncertainty estimates in the stratosphere. Around the tropopause and below, the total TROPOMI

uncertainty is on average underestimated by a factor of two at maximum.

The information content of the operational TROPOMI ozone profile retrieval is assessed through the analysis of its averaging kernels. Although each ozone profile is characterised by about five to six independent pieces of information (DFS), it must be





kept in mind that these are not equally distributed over the derived product consisting of six subcolumns. The kernel matrix row sums reveal a vertical sensitivity close to 100 % at all altitudes from about 20 to 50 km yet decreasing rapidly above and below.

Towards the surface, on average 50 % of the retrieved information originates from the prior profile. The corresponding kernel peaks at about 10 km altitude on average. Typically, the kernel peak (information barycentre) only lies at the nominal retrieval altitude where the sensitivity approaches unity. As another measure of the vertical sounding accuracy, the effective vertical resolution of the profile retrieval usually ranges within 10-15 km, with a minimum close to 7 km in the middle stratosphere, which is below the retrieval grid resolution (6 km at maximum). The higher sensitivities and effective vertical resolutions are

typically observed for longer atmospheric optical paths, i.e., for solar zenith angles above $60°$.

The path length dependence of the retrievals' information content also translates into seasonal and meridian dependencies of the—especially tropospheric—bias and affects the oscillations in the stratospheric bias. A similar but reduced effect can be seen for the viewing zenith angle. Additionally, the bias is negatively correlated with the surface albedo for the 6-12 km subcolumn, despite the latter's apparently slightly positive correlation with the retrieval degrees of freedom. For the lowest retrieval levels

(0-6 km), this correlation looks somewhat compensated by the increased atmospheric penetration of the sunlight at low solar zenith angles (0-30°). On the other hand, one can also observe lower-DFS profiles with nearly-zero surface sensitivity in combination with a highly overcompensating sensitivity around the UTLS, ranging up to three and above. These retrievals occur for scenes that have both high SZA and high surface albedo, mostly around the Antarctic.

The five years of TROPOMI ozone profile data under study show a slight DFS degradation throughout the mission (next to a

jump from the ground pixel resolution change). Comparisons with the ozonesonde data reveal significant positive drifts near 2 %/year in the tropics and mid-latitudes from the surface to the UTLS, while 1-2 % per year negative drifts are observed above. This makes the current operational TROPOMI ozone profile product and its subcolumn derivatives unsuitable for vertically resolved trend studies. However, no significant drift is detected for the vertically integrated profile. This agrees with the operational TROPOMI total ozone column retrieval, although the latter is consistently about 5 % higher than the integrated ozone

profile.

Four scientific TROPOMI ozone profile retrieval algorithms have been found described in the literature. Two of these also provide joint retrievals with the infrared CrIS instrument that orbits three minutes ahead of S5P. Bremen University's TOPAS product performs slightly better (showing a vertically consistent 5 % bias) than the operational one in the stratosphere, while NASA's MUSES algorithm shows total tropospheric uncertainties below 20 %. Apart from these exceptions, the operational

ozone profile product demonstrates a comparable or lower uncertainty than the scientific products. It is moreover found to be at least partially compliant with all mission requirements. The TROPOMI instrument as such provides a crucial component to the present-day global ozone observing system. It provides important contributions to, amongst others, the Global Climate Observing System (GCOS), ESA's Climate Change Initiative on ozone (and its Climate Space follow-up), the Copernicus Climate Change Service (C3S), the IGAC Tropospheric Ozone Assessment Report-II (TOAR-II), and several activities endorsed

by the Committee on Earth Observation Satellites (CEOS).



*Data availability.* Sentinel-5 Precursor TROPOMI data are available from the Copernicus Open Access Hub (https://scihub.copernicus.eu). This data is open for use by the public, subject to the data policy. The ozonesonde and lidar data used in this publication were obtained as part of the Network for the Detection of Atmospheric Composition Change (NDACC, https://ndacc.org), the World Ozone and UV Radiation Data Centre (WOUDC, https://woudc.org/), the Tropospheric Ozone Lidar Network (TOLNet, https://www-air.larc.nasa.gov/missions/TOLNet/), NASA's Southern Hemisphere ADditional OZonesonde programme (SHADOZ, https://tropo.gsfc.nasa.gov/shadoz), and NOAA's Global Monitoring Laboratory (GML, https://gml.noaa.gov/). They are publicly available through the respective network data archives and partially—yet including harmonization to the GEOMS format and quality control—through ESA's Validation Data Centre (EVDC, https://evdc.esa.int/).

## Appendix A:  Detailed tables and figures

*Author contributions.* AK and DH conceived, coded, performed, and wrote the validation analysis initiated and coordinated by JCL. SDP and PV developed, implemented, and described the retrieval algorithm, heavily supported by MS, JDH, and MTL. TL coordinated the inclusion of the tropospheric lidar data and their description. SC, TV, JG, JCL, and ON contributed to the data processing and discussion at all stages of the validation analysis. AMF and IB coordinate and maintain the ozonesonde data collection at EVDC, with additional data processing tools provided by SN. RVM, HS, VD, SGB, BJJ, WS, DT, DEK, RMS, and AMT maintain and provide access to the underlying ozonesonde and lidar data archives. CZ and AD manage the Copernicus S5P mission, the ESA/COpernicus ATM-MPC and the S5PVT. All authors revised and commented on the manuscript.

*Competing interests.* At least one of the (co-)authors is a member of the editorial board of Atmospheric Measurement Techniques.

*Acknowledgements.* Part of the reported work was carried out in the framework of the ESA/Copernicus Atmospheric Mission Performance Cluster (ATM-MPC), contracted by the European Space Agency (ESA/ESRIN, Contract No. 4000117151/16/I-LG) and supported by the Belgian Federal Science Policy Office (BELSPO), the Royal Belgian Institute for Space Aeronomy (BIRA-IASB), the Netherlands Space Office (NSO), and the German Aerospace Centre (DLR). Part of this work was also supported by the S5P Validation Team (S5PVT) AO project CHEOPS-5p (ID #28587, Co-PIs J.-C. Lambert and A. Keppens, BIRA-IASB) funded by the BELSPO ProDEx project TROVA-E2 (PEA 4000116692). The authors express special thanks to Olivier Rasson for satellite and ground-based data post-processing. This work contains modified Copernicus Sentinel-5 Precursor satellite data (2018–2023) processed by KNMI and post-processed by BIRA-IASB. This work also contains modified GOME-2B and OMI satellite data processed by BIRA-IASB and DLR in the framework of ESA's Ozone_cci+ project (https://climate.esa.int/en/projects/ozone/) and post-processed by BIRA-IASB. The instrument PIs and staff at the lidar and ozonesonde stations (with their affiliations identified in the Appendix tables) are warmly thanked for their sustained effort on maintaining high quality measurements and for valuable scientific discussions. ESA and NILU are acknowledged for facilitating operational access to ground-based validation data through the ESA Atmospheric Validation Data Centre (EVDC).





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



**Table 1.** Important changes in the operational TROPOMI ozone profile version 2.4.0 processing of near real-time (NRTI), offline (OFFL), and reprocessed (RPRO) data. The difference between the two soft-calibration versions is that for v1 only DDS6 (sixth diagnostic data set) orbits were used to compute the correction parameters, while for v2 a combination of DDS6 and OFFL orbits was used.

| Date | Operational processing change |
|---|---|
| 2018/04/30 | RPRO start date using soft-calibration v2 |
| 2019/08/06 | TROPOMI pixel resolution change |
| 2022/07/25 | RPRO end date |
| 2022/07/25 | OFFL start date using soft-calibration v1 |
| 2023/01/15 | OFFL soft-calibration update to v2 |
| 2023/03/15 | NRTI checkerboard pattern (v2.5.0) |





**Table 2.** TROPOMI mission requirements for the operational ozone profile product, and its current compliance. Sub-column requirements (above the middle bar) originate from the Science Requirements Document (2008), while profile requirements (below the middle bar) originate from the Geophysical Validation Requirements document (2014).

| Requirement | Compliance |
|---|---|
| 0-6 km accuracy $\leq 20$ % | Fully, but typically $\geq 50$ % of the retrieved profile originates from the prior. |
| 6-12 km accuracy $\leq 12$ % | Fully. |
| 12-18 km accuracy $\leq 5$ % | Fully. |
| 18-50 km accuracy $\leq 3$ % | Partially, due to vertical oscillations in the bias, reaching 15 %. |
| Vertical resolution $\leq 6$ km | Partially, as the vertical grid complies, but not the effective vertical resolution measured from the AKs. |
| Systematic uncertainty $\leq 30$ % | Fully, actually below 15 %. |
| Random uncertainty $\leq 10$ % | Partially, order of 30 % below the tropopause, order of 10 % or lower above. |





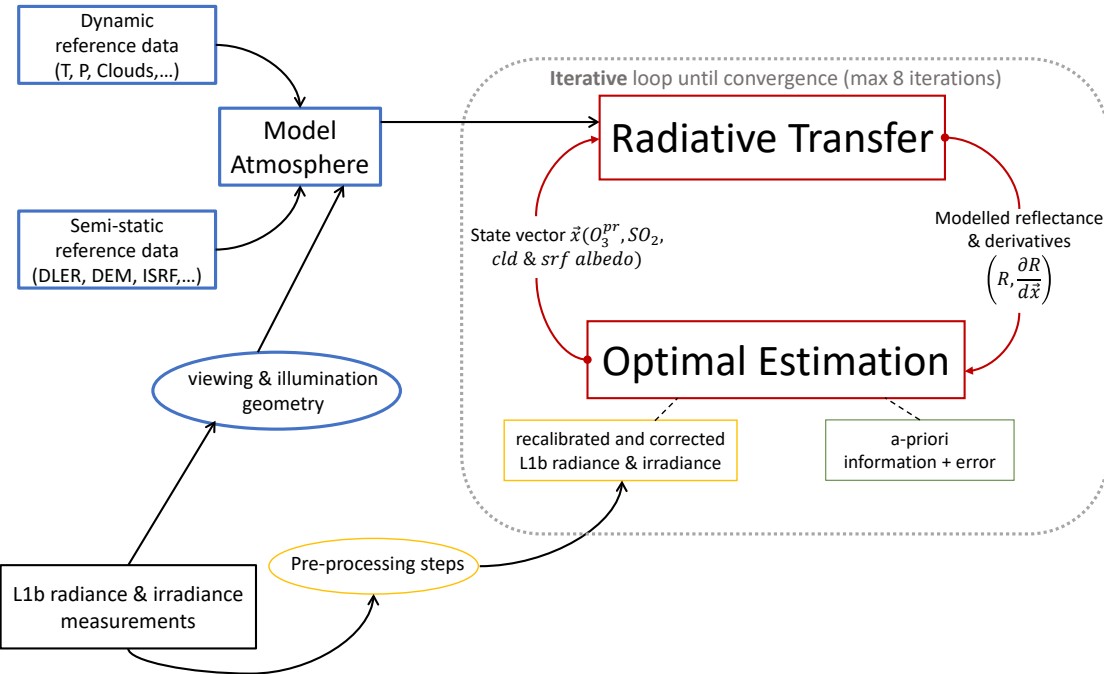

**Figure 1.** Schematic overview of the ozone profile retrieval algorithm. The main elements are the Radiative Transfer calculations (the forward model calculations) which provide the input modelled reflectance and its derivative to the Optimal Estimation method, which calculates the state vector with respect to the fitted parameters in the parenthesis. An iterative loop between these two elements (the light gray contour) is performed until convergence is reached.



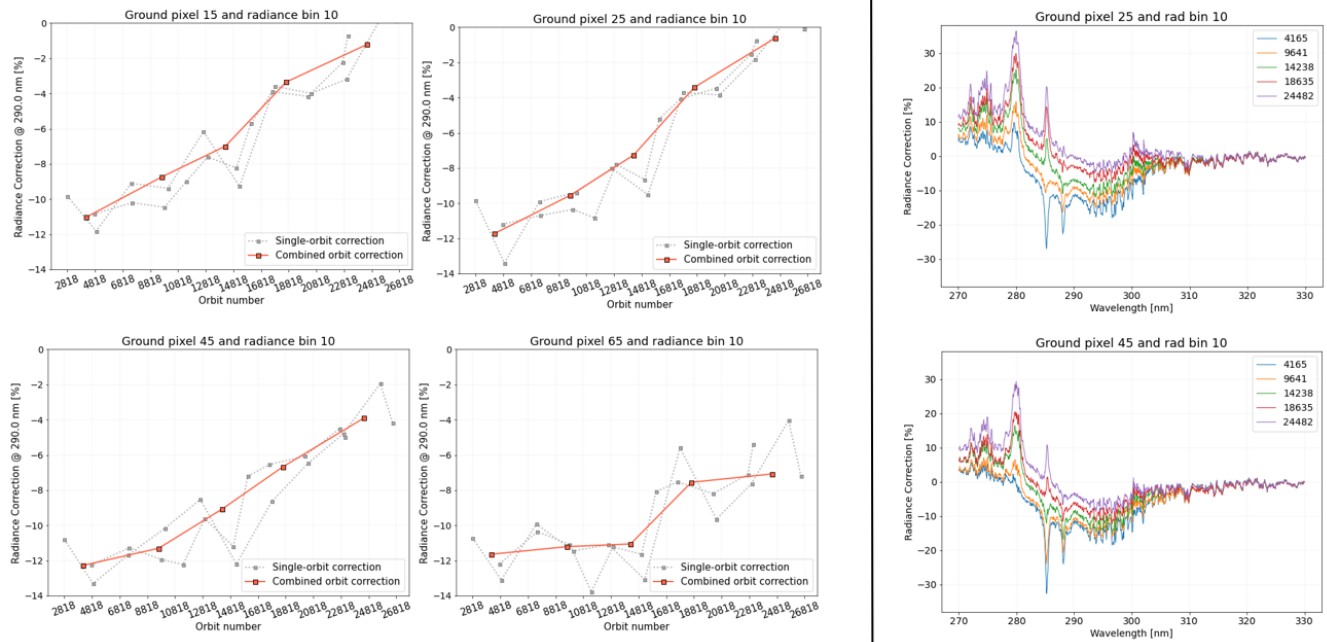

**Figure 2.** The yearly radiometric correction implemented from the beginning of the TROPOMI mission in 2018 up to December 2022 is shown in the left plot for four ground pixels (15, 25, 45 and 65), the middle radiance bin, and at a specific wavelength (290 nm). On the right, the radiometric correction is shown per year as a function of the wavelength, for two ground pixels (25 and 45) and the middle radiance bin, with the corresponding orbit numbers shown in the legend.





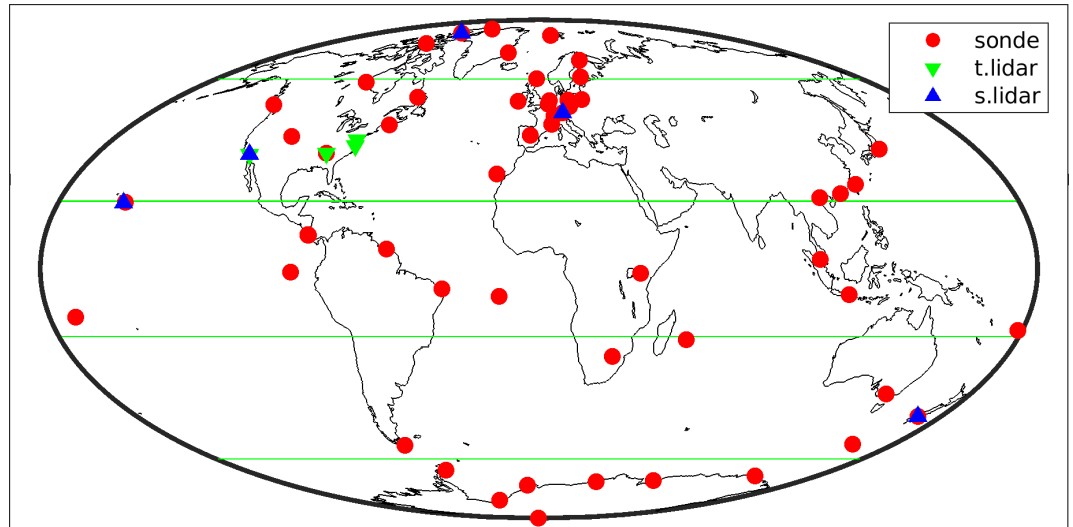

**Figure 3.** Geographical distribution of the ozonesonde, tropospheric (t.) lidar, and stratospheric (s.) lidar stations having coincident measurements with the S5P ozone profile data for the comparisons reported in this work. Horizontal green lines separate the five latitude zones that are distinguished in the comparative analysis with respect to ozonesondes.



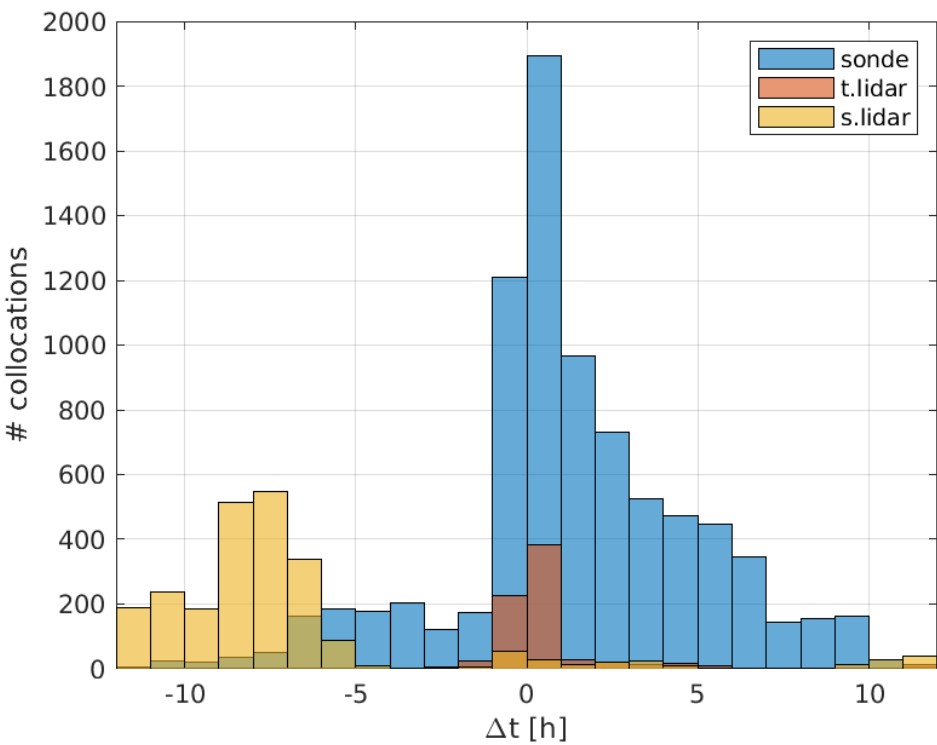

**Figure 4.** Histogram for the time differences (in hourly bins) between coincident TROPOMI and reference measurements: ozonesondes in blue, tropospheric (t.) lidar in red, and stratospheric (s.) lidar in yellow. All bins add up to the collocation numbers in Figure 11. Positive values indicate TROPOMI measured later than the reference instrument.







**Figure 5.** The mean ozone content in the six sub-columns of the ozone profile for the month of October 2020, during the ozone hole condition. The first sub-column at the bottom of the column starts at the bottom left and the last sub-column is at the top right. The colour scale is optimized per each sub-column.



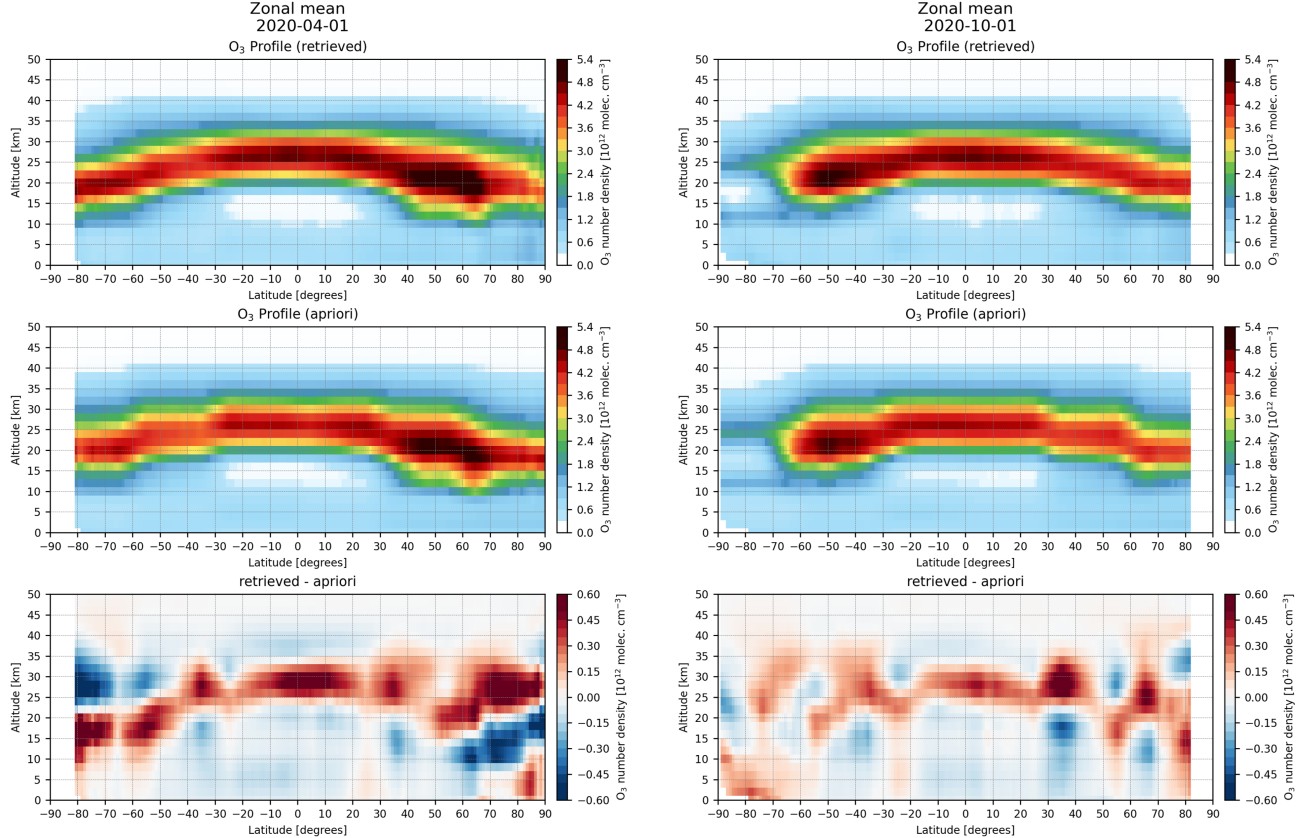

**Figure 6.** Zonal average of the ozone profile retrieval (top panel), the a-priori ozone profile (middle panel) and their relative difference (bottom plot, retrieval minus prior) for two days in 2020, April 1st (left column) and October 1st (right column).



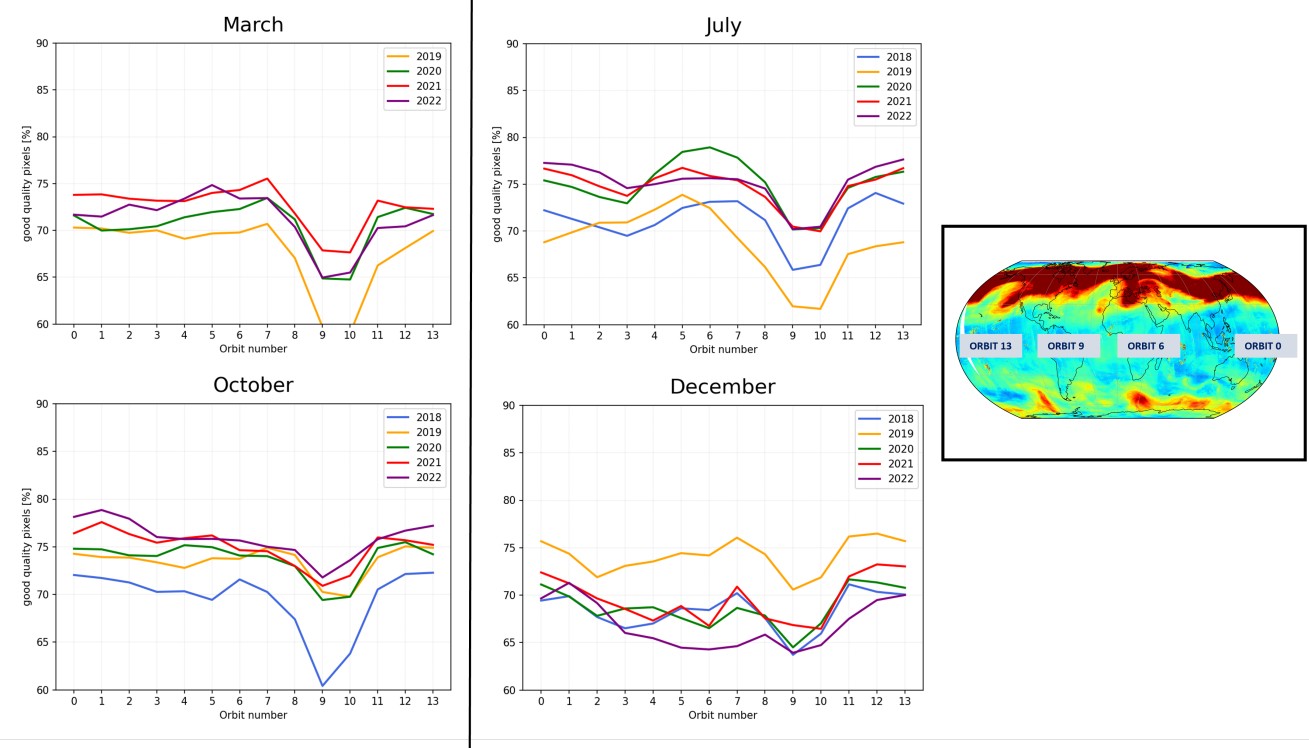

**Figure 7.** The percentage of good quality pixels after applying the recommended filters on the QA value, cost function and strongly deviating from the a-priori condition in Eq. (2). Each line represents the average percentage of the same orbit number over all the days of the specific month analysed. In the right, the order of the orbits number is shown.



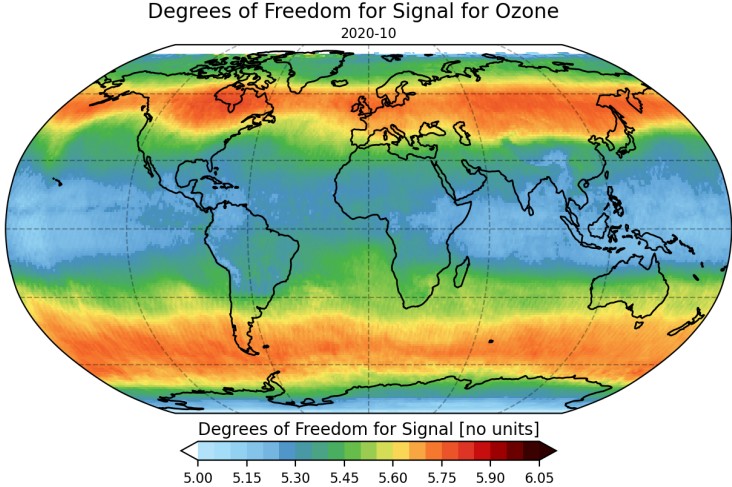

**Figure 8.** The monthly mean of the ozone degrees of freedom in October 2020.

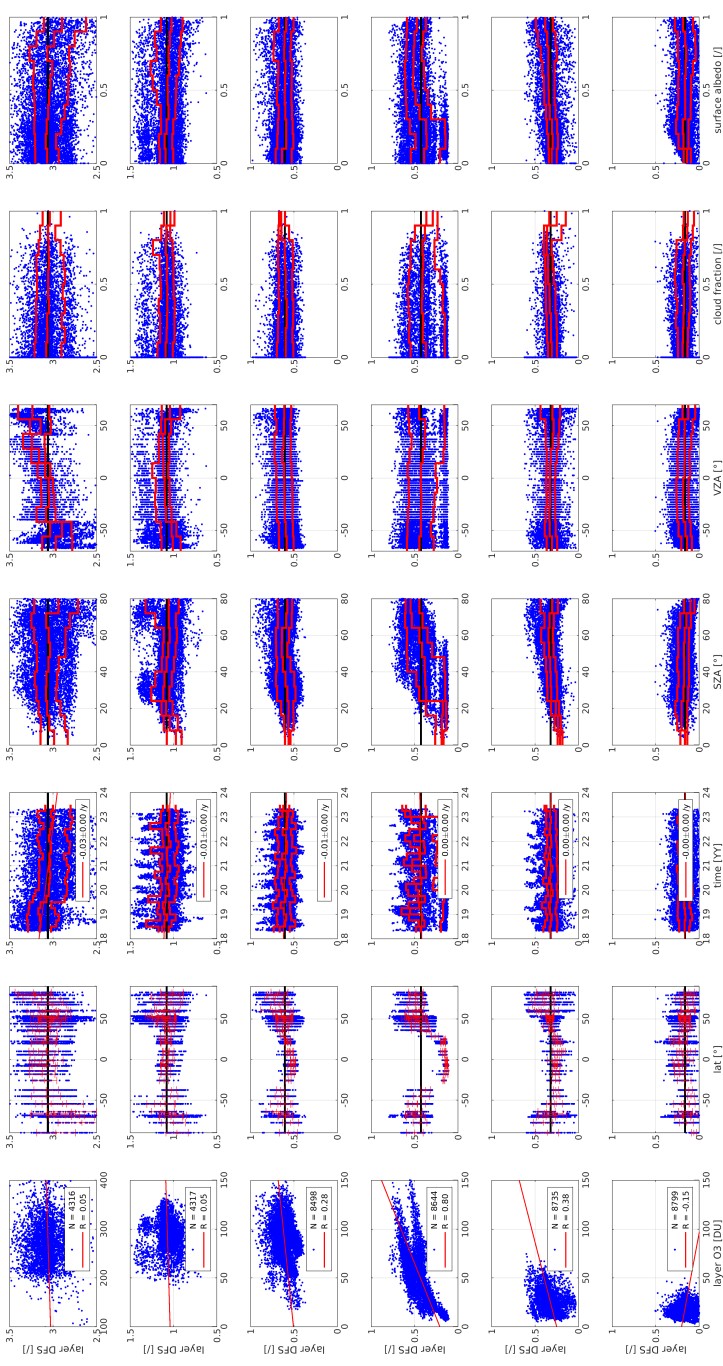

**Figure 9.** Correlation (R) between the six subcolumn DFS values and the six ozone subcolumns (bottom to top row: 0-6 km, 6-12 km, 12-18 km, 18-24 km, 24-32 km, 32-82 km) for N retrievals by TROPOMI (first column in landscape view). Subsequent columns show the layer-DFS as a function of latitude, time, solar zenith angle (SZA), viewing-zenith angle (VZA), cloud fraction, and surface albedo. 84, 50, and 16 % quantiles are added in red (per station, per season, or for ten bins), together with the overall layer mean (black line). Yearly drifts are added to the temporal dependence plots.

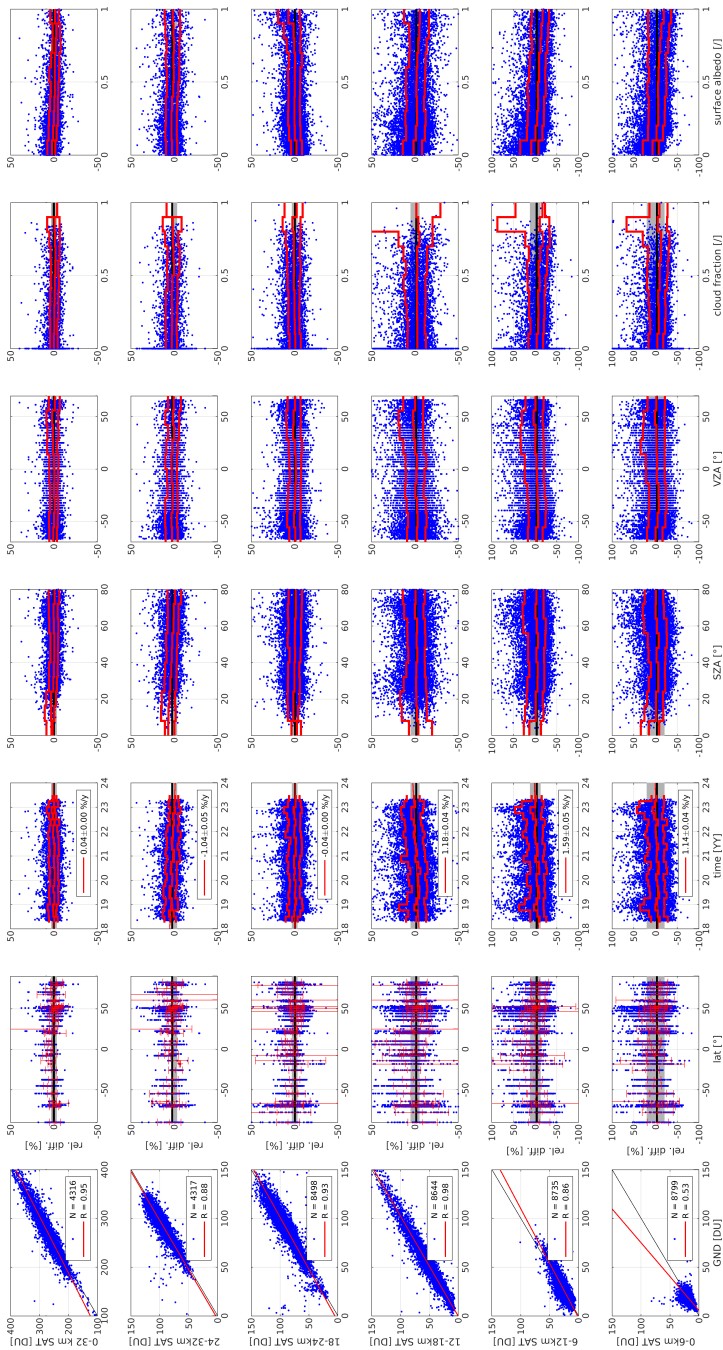

**Figure 10.** Correlation (R) between N lowest five ozone subcolumns as observed by TROPOMI and the coincident vertically integrated ozonesonde measurements (first column in landscape view), and their overall sum (top row). Subsequent columns show the differences between satellite and ozonesonde subcolumns as a function of latitude, time, solar zenith angle (SZA), viewing-zenith angle (VZA), cloud fraction, and surface albedo. 84, 50, and 16 % quantiles are added in red (per station, per season, or for ten bins), together with the overall mean difference (black line), and the product requirements for each subcolumn (grey areas). Yearly drifts are added to the temporal dependence plots.

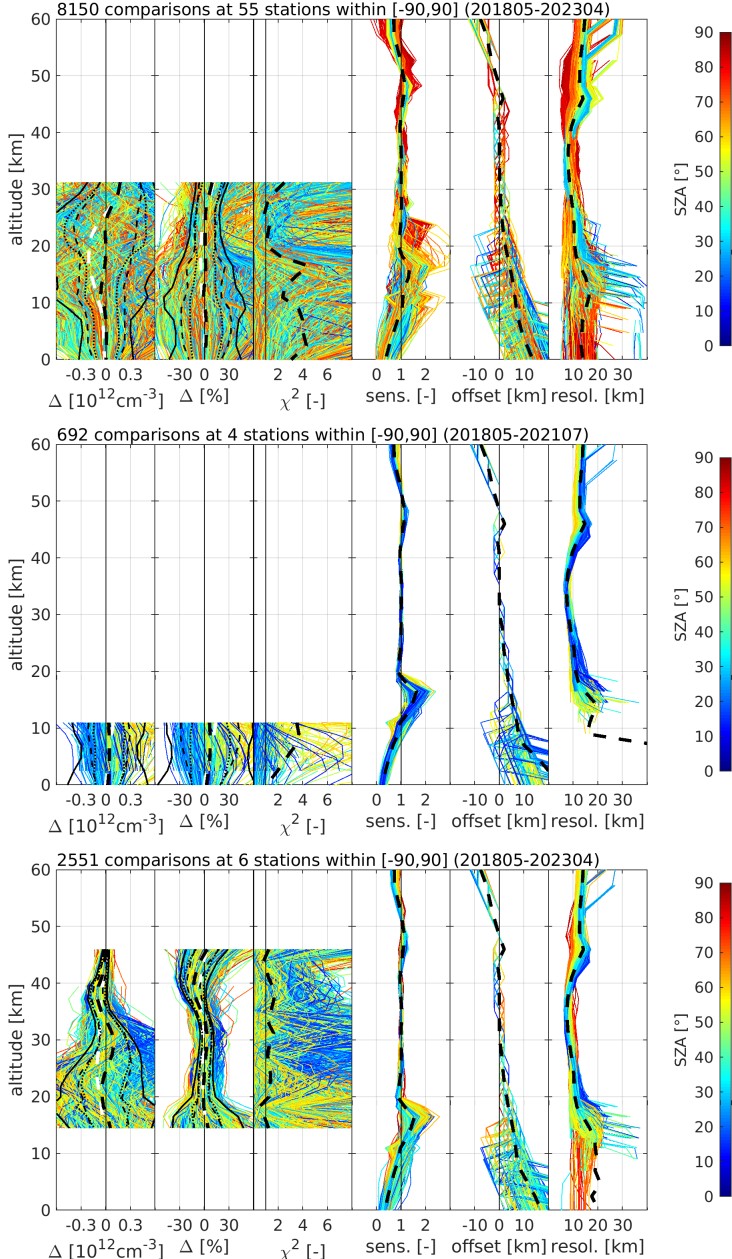

**Figure 11.** Comparison between S5P RPRO/OFFL ozone number density profile data and all co-located ground-based reference measurements. Every subplot shows six graphs, respectively, from left to right: the difference and the percent relative difference between S5P and ozonesonde (top), tropospheric lidar (middle), or stratospheric lidar (bottom), the chi-square profile, the vertical sensitivity, the altitude registration offset, and the averaging kernel FWHM associated with the satellite retrieval. The colour scale indicates the TROPOMI solar zenith angle. Black dashed lines show mean values (thick lines) and standard deviations (thin lines, around the mean), while white dashed lines indicate the mean difference between the a-priori profile and the reference measurement. Dotted black lines indicate the total ex-ante (inductive) uncertainty of TROPOMI and the reference measurements combined (around the mean difference). The black full lines show the same, after adding the retrieval's smoothing error.





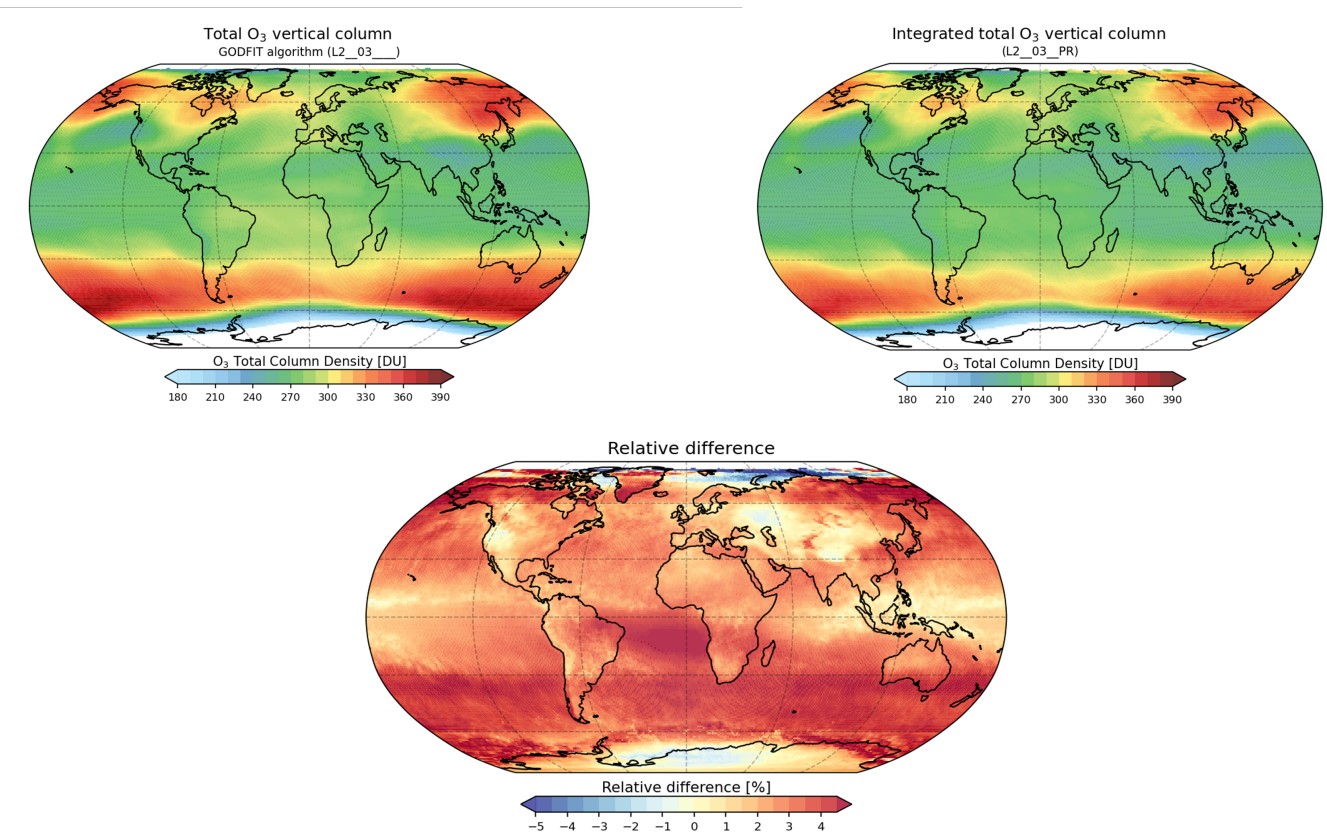

**Figure 12.** Relative difference (bottom) between the total ozone column of the operational TROPOMI GODFIT algorithm (L2__O3____), top left, and the integrated operational ozone profile product, top right, for October 2020.





**Table A1.** List of northern hemisphere ozonesonde stations providing correlative data for the TROPOMI validation in this work. Subsequent columns show the station name, location, archive / PI institute abbreviation, validation period (Full for the entire five year TROPOMI time range: May 2018 to April 2023), and number of co-locations in the comparative validation. Horizontal black lines separate the latitude zones defined in Figure 3.

| Name | Location [°N,°E] | Archive / PI inst. | Period [YY/MM] | Co-locs. |
|---|---|---|---|---|
| Alert | [82.45, -62.51] | WOUDC / MSC | 18/05 - 22/06 | 91 |
| Eureka | [79.99, -85.93] | WOUDC / MSC | 18/05 - 23/03 | 151 |
| Ny-Alesund | [78.93, 11.93] | NDACC / AWI | 18/05 - 23/03 | 123 |
| Resolute | [74.72, -94.98] | NDACC / MSC | 18/05 - 20/04 | 56 |
| Scoresbysund | [70.48, -21.95] | NDACC / DMI | Full | 174 |
| Sodankyla | [67.37, 26.63] | NDACC / FMI | Full | 107 |
| Jokioinen | [60.81, 23.50] | NDACC / FMI | 20/01 - 22/04 | 9 |
| Lerwick | [60.13, -1.18] | NDACC / UKMO | 18/05 - 22/09 | 201 |
| Churchill | [58.74, -94.07] | NDACC / MSC | 19/01 - 22/02 | 103 |
| Goose Bay | [53.31, -60.36] | NDACC / MSC | 18/05 - 22/06 | 126 |
| Legionowo | [52.40, 20.97] | NDACC / IMGW | Full | 238 |
| Lindenberg | [52.21, 14.12] | NDACC / DWD | Full | 245 |
| De Bilt | [52.10, 5.18] | NDACC / KNMI | 18/05 - 23/03 | 195 |
| Valentia | [51.94, -10.25] | NDACC / ME | Full | 128 |
| Uccle | [50.80, 4.35] | NDACC / RMI | Full | 659 |
| Port Hardy | [50.68, -127.38] | WOUDC / MSC | 18/06 - 23/03 | 117 |
| Praha | [50.00, 14.44] | NDACC / CHMI | 19/01 - 23/04 | 234 |
| Hohenpeissenberg | [47.81, 11.01] | WOUDC / DWD | Full | 626 |
| Payerne | [46.82, 6.95] | WOUDC / MeteoSwiss | Full | 548 |
| Obs. Haute Provence | [43.94, 5.71] | NDACC / LATMOS | 18/05 - 22/11 | 189 |
| Yarmouth | [43.87, -66.11] | WOUDC / MSC | 18/05 - 22/06 | 186 |
| Madrid | [40.45, -3.72] | WOUDC / INM | Full | 236 |
| Boulder | [40.03, -105.25] | NOAA / GML | 23/03 | 1 |
| Tsukuba | [36.05, 140.13] | WOUDC / JMA | Full | 210 |
| Huntsville | [34.72, -86.64] | NOAA / UAH | 18/05 - 20/03 | 22 |
| Izana | [28.31, -16.50] | NDACC / AEMET | 18/05 - 22/12 | 182 |
| Taipei | [25.04, 121.51] | WOUDC / CWBT | 18/11 - 22/03 | 17 |
| Hong Kong Obs. | [22.31, 114.17] | WOUDC / HKO | 18/05 - 22/06 | 183 |
| Hanoi | [21.03, 105.85] | SHADOZ / GSFC | 18/05 - 21/11 | 84 |
| Hilo | [19.72, -155.07] | NOAA / GML | Full | 237 |
| Costa Rica (3 locs.) | [9.94, -84.04] | SHADOZ / NCAR | 18/05 - 23/03 | 112 |
| Paramaribo | [5.81, -55.22] | NDACC / KNMI | Full | 161 |
| Sepang Airport | [2.73, 101.70] | SHADOZ / MMD | 18/05 - 22/12 | 79 |





**Table A2.** List of southern hemisphere ozonesonde stations providing correlative data for the TROPOMI validation in this work. Subsequent columns show the station name, location, archive / PI institute abbreviation, validation period (Full for the entire five year TROPOMI time range: May 2018 to April 2023), and number of co-locations in the comparative validation. Horizontal black lines separate the latitude zones defined in Figure 3.

| Name | Location [°N,°E] | Archive / PI inst. | Period [YY/MM] | Co-locs. |
|---|---|---|---|---|
| San Cristobal | [-0.92, -89.60] | SHADOZ / USFQ | 21/12 - 22/12 | 23 |
| Nairobi | [-1.27, 36.80] | SHADOZ / MeteoSwiss | 18/06 - 20/03 | 43 |
| Ascension Island | [-1.98, -14.42] | SHADOZ / GSFC | 18/05 - 22/09 | 144 |
| Natal | [-5.83, -35.20] | SHADOZ / INPE | 18/05 - 22/12 | 83 |
| Watukosek | [-7.50, 112.60] | SHADOZ / BRIN | 21/03 - 22/12 | 23 |
| Pago Pago | [-14.23, -170.56] | NOAA / GML | Full | 155 |
| Suva | [-18.13, 178.40] | NOAA / GML | Full | 54 |
| St. Denis | [-20.90, 55.48] | SHADOZ / LACY | 18/05 - 20/12 | 87 |
| Irene | [-25.92, 28.22] | SHADOZ / SAWS | 18/05 - 22/08 | 33 |
| Broadmeadows | [-37.69, 144.95] | WOUDC / ABM | 18/05 - 23/02 | 226 |
| Lauder | [-45.04, 169.68] | NDACC / NIWA | 18/05 - 22/06 | 208 |
| Macquarie | [-54.50, 158.95] | WOUDC / ABM | 18/05 - 23/02 | 231 |
| Ushuaia | [-54.85, -68.31] | WOUDC / SMNA | 18/08 - 19/11 | 18 |
| Marambio | [-64.23, -56.62] | WOUDC / FMI | 18/08 - 19/11 | 43 |
| Dumont d'Urville | [-66.67, 140.02] | NDACC / LATMOS | 18/08 - 19/12 | 13 |
| Davis | [-68.58, 77.97] | WOUDC / ABM | 18/08 - 23/02 | 166 |
| Syowa | [-69.01, 39.58] | WOUDC / JMA | 18/08 - 23/04 | 176 |
| Neumayer | [-70.68, -8.26] | NDACC / AWI | 18/08 - 23/03 | 209 |
| Belgrano | [-77.77, -38.18] | NDACC / INTA | 18/10 - 23/02 | 52 |
| Amundsen Scott | [-89.98, -24.80] | NOAA / GML | 18/10 - 21/11 | 72 |



**Table A3.** List of DIAL stations providing correlative data for the TROPOMI validation in this work. Subsequent columns show the station name, location, archive / PI institute abbreviation, validation period (Full for the entire five year TROPOMI time range), and number of co-locations in the comparative validation. The horizontal black line separates tropospheric (top) and stratospheric (bottom) lidar instruments.

| Name | Location [°N,°E] | Archive / PI inst. | Period [YY/MM] | Co-locs. |
|---|---|---|---|---|
| Greenbelt | [38.99, -76.84] | TOLNet / GSFC | 20/05 - 21/07 | 118 |
| Langley | [37.10, -76.39] | TOLNet / LaRC | 18/05 - 19/11 | 15 |
| Huntsville | [34.72, -86.64] | TOLNet / UAH | 18/05 - 20/08 | 103 |
| Table Mountain (t.) | [34.40, -117.70] | TOLNet / JPL | 18/05 - 20/07 | 456 |
| Eureka | [79.99, -85.93] | NDACC / U. Toronto | 20/03 | 2 |
| Hohenpeissenberg | [47.81, 11.01] | NDACC / DWD | Full | 532 |
| Obs. Haute Provence | [43.94, 5.71] | NDACC / LATMOS | Full | 595 |
| Table Mountain (s.) | [34.40, -117.70] | NDACC / JPL | Full | 722 |
| Mauna Loa | [19.53, -155.58] | NDACC / JPL | 18/05 - 22/11 | 592 |
| Lauder | [-45.04, 169.68] | NDACC / NIWA | 18/05 - 21/07 | 108 |





**Figure A1.** The percentage of good quality pixels after applying separately each of the recommended filters on the QA value, cost function and strongly deviating from the a-priori condition (2). Each line represents the average percentage of the same orbit number over all the days of the specific month analysed.



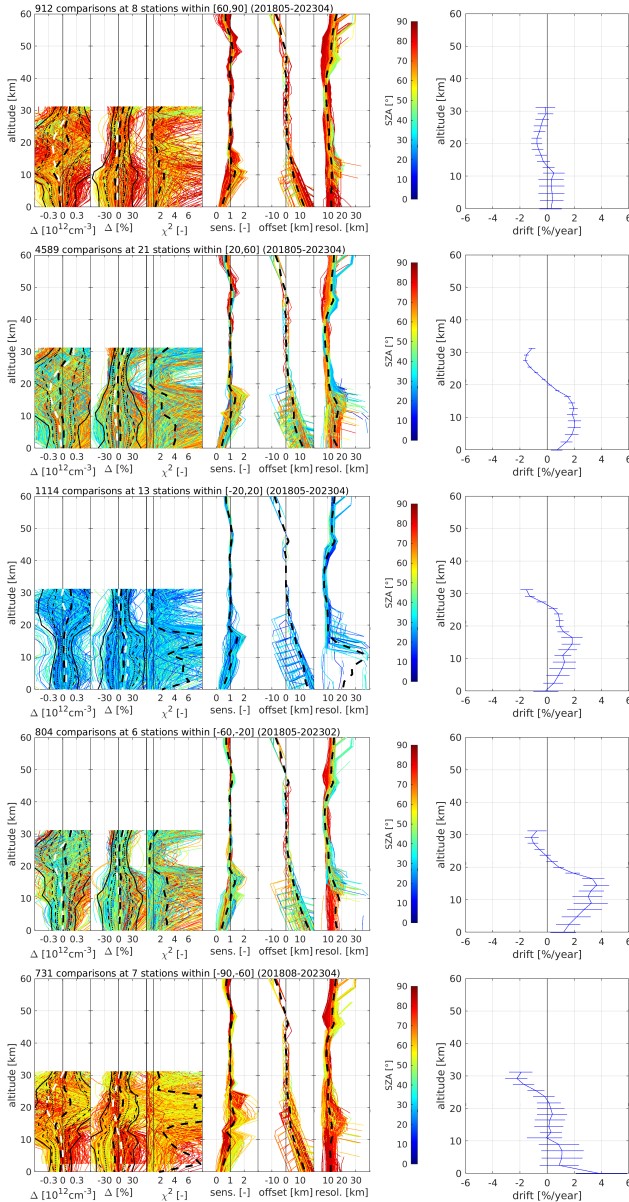

**Figure A2.** Comparison between S5P RPRO/OFFL ozone number density profile data and all co-located ozonesonde reference measurements, for five latitude bands sorted north to south. Every left subplot shows six graphs, respectively, from left to right: the difference and the percent relative difference between S5P and ozonesonde, the chi-square profile, the vertical sensitivity, the altitude registration offset, and the averaging kernel FWHM associated with the satellite retrieval. The colour scale indicates the TROPOMI solar zenith angle. Black dashed lines show mean values (thick lines) and standard deviations (thin lines, around the mean), while white dashed lines indicate the mean difference between the a-priori profile and the reference measurement. Dotted black lines indicate the total ex-ante (inductive) uncertainty of TROPOMI and the reference measurements combined (around the mean difference). The black full lines show the same, after adding the retrieval's smoothing error. Plots on the right show altitude-dependent yearly drifts for the same ozonesonde comparisons, with two-sigma horizontal error bars resulting from a bootstrapping with 1000 samples.