# Peer review of "Five years of Sentinel-5p TROPOMI operational ozone profiling and geophysical validation using ozonesonde and lidar ground-based networks"

_Atmospheric Measurement Techniques, 2023_

## Referee Comment (RC1)

**Overall Comments:**

Keppens et al. provide a comprehensive description of TROPOMI operational ozone retrievals and retrieval characteristics (sensitivity, smoothing error, averaging kernels), along with validation results spanning five years of their product. In general, I recommend this manuscript for publication after the following comments being addressed.

Both the retrieval (Section 2) and validation procedures (Section 3) outlined in this paper are already part of the operational process. Therefore, most of my major concerns are focused on the validation results (Section 4).

1. From Figure 12 (relative differences of total columns between S5P products), I think, it is of importance to dig up the sources of differences in the western ocean out of south Africa. It could be either S5P profile measurements miss the biomass-burning signals due to low sensitivity or S5P total ozone retrievals are contaminated. Please give more descriptions.

[Figure]

2. Figure 5: the geophysical distribution of the mean sub-column layers looks very informative for both the lower troposphere and upper stratosphere, showing no apparent artifacts. However, S5P product is expected to offer high spatiotemporal information compared to other processors (e.g., OMI, GOME). Therefore, I highly encourage to provide an example of the daily (or a few days average) tropospheric ozone map zoomed in on a specific continent (e.g., Europe, Asia).

3. I would like to recommend moving Section 2.3 to Section 3 as a part of validation meteorology, rather than the retrieval meteorology.

4. Line 224 What is the threshold value applied for the quality of fit (cost function) in data screening?

5. Line 229-253: the description of the operational validation system is better to be placed in Appendix to ensure the consistency and conciseness in text.

6. This manuscript should be revised overall to enhance readability.
   Even when the full name of the term is occasionally abbreviated in this paper, there is frequent inconsistency in the use of abbreviations and full names, even when the abbreviation is more familiar to the reader than the full name. DFS, TOLNet, WOUDC, NDACC, AK, FWHM, GAW, SHADOZ.

7. Section 3.1 should be placed just before Section 3.2.4 or merged into Section 3.2.4 to ensure the consistency and conciseness in text. And, it is much better to re-organize Section 4 into three parts, Retrieval characteristics including all AK, DFS, FWHM, sensitivity…, Geophysical distribution and validation results (comparison results between S5P and ground reference, between S5P profiles and S5P column). Correspondingly, Figures 9-11 should be re-organized to enhance the readability.

8. Multi-Figures should have captions (Figure 1.a, 2.b, 3.b).

9. Figure 9 and Figure 11. It is not necessary to perform the retrieval characteristics (DFS, sensitivity, FWHM, offset) at specific stations and specific reference dataset. I think, a few orbit files are enough to specify the dependence of the retrieval characteristics on the geophysical parameters (SZA, VZA, cloud fraction…).

10. This paper strongly assured that the impact of sampling and smoothing errors on comparison results between S5P and reference are insignificant, thanks to the application of tight co-location criteria and AK smoothing of the reference observations. But, the substantial offset between nominal retrieval altitude and effective vertical retrieval altitude could introduce artificial features, right?

11. Please add legends in Figure 11 for many lines (dashed, dotted, thin). And need to revise Figure 11. It is hard to draw any insight on the data assessments as a function of SZA with the current way to put all individual profiles with different color-cording as a function of SZA. Maybe take a look at the mean difference/standard deviations for several SZA regimes (SZA < 40, SZA all, SZA > 60….)

12. In Section 4.7 and the conclusions (pages 585-595), the authors present a comparative analysis of data quality, comparing their TROPOMI operational ozone profiles with those from other research products (Zhao et al. 2020, Mettig et al. 2021; 2022, Malina et al., 2020), relying on literature assessment reports. Specifically, they assert that, "Apart from these exceptions (Malina et al., 2020 and Metting et al. products), the operational product demonstrates comparable or lower uncertainty than the scientific products." However, it is not proper to draw definitive conclusions without conducting cross-validation using the same reference and validation criteria. With the difficulty in collecting other S5P ozone profile products for an long-term period, I recommend removing Section 4.7, except for the comparison results between S5P total column and S5P ozone profile-integrated column. Instead, provide a concise summary of other scientific products developed for delivering ozone profile information from TROPOMI measurements in the Introduction.

13. Figure 2: what does '" the middle radiance bin mean"? does it indicate the middle spectral pixel of 270-330 nm, at 290 nm? if so, please delete the middle radiance bin in part of describing Figure for the right panel of Figure 2.

14. Line 545 The main elements of the operational retrieval algorithm include the spectral pre-processing, which involves spectral/spatial regridding and wavelength/radiometric correction, a forward model, and an optimal estimation based inverse model.

15. Line 553 The reference dataset used here includes WOUDC ozonesondes, TOLN tropospheric lidars, and NDACC stratospheric lidars.

16. Line 508 delete "additionally applied to the MLS and to OMPS for intercomparison", And, connecting to the following sentence, like, "which applied for joint UV-IR retrievals from TROPOMI and CrIS.

17. Line 515 delete ",which is applied ~ data"

18. C17. Line 547 ranging from ➔ spanning

19. Line 580. Please specify "Observed above" e.g. observed for the stratospheric ozone retrievals.

20. Line 584 Please provide any reference to "this agrees with the operational TROPOMI total ozone column retrieval". And, the long-term stability is commonly assured for other S5P L2 products?

---

## Referee Comment (RC2)

**Referee report to the "Five years of Sentinel-5p TROPOMI operational ozone profiling and geophysical validation using ozonesonde and lidar ground-based networks" manuscript by Arno Keppens et al.**

The manuscript describes the retrieval algorithm used for the operational retrieval of ozone vertical profiles from Sentinel-5p TROPOMI measurements. Sensitivity of the retrievals and uncertainties of the results are well described. The retrieved ozone profiles are validated using the ozone sonde and lidar measurements. The topic of the manuscript is of a high scientific importance for ozone data users and is well suited for publication in AMT. In the first part of the manuscript authors describe the instrument, algorithm, data processing chain, quality assurance methods and reference data in much detail, very concise and clear. This part of the manuscript is written almost perfectly. Unfortunately, in Section 4, which describes the retrieval sensitivity and validation results, authors decided to save the space and joined tens of panels with thousands of curves into single plots. This make the plots absolutely unreadable (Figs 9, 10, 11 and A2). Although the text is still good written, the reader cannot follow the discussion and verify the conclusions of the authors because no information can be read from the plots. In my opinion, the authors have to reconsider the way how they present their results in Section 4 to make the manuscript suitable for the publication. It should be analyzed which information is important and how to present it without making the plots looking like squeezed together unresolved color spots. A rotation of the plots by 90 degrees is also not a really good idea. This makes the reader terribly difficult to look at the plots when reading the text.

**Detailed comments**

- The paragraph mentioning previous retrievals (lines 505–516, Section 4.7) should be moved to the introduction.

- Figure 1: the right box shows an endless loop between the radiative transfer and optimal estimation.

- Line 125: for the pixel size after August 6, 2019 please indicate which size is cross-track and which along-track.

- Line 147: Usage of the CAMS data, with assimilated MLS profiles, and scaling them to OMPS total columns is expected to give a very good approximation for the ozone profile. In this respect author should put a bit more focus to show that TROPOMI measurements increase the information content in comparison to a priori. Maybe it is already shown in plots in Section 4, they are, however, completely unreadable for me.

- Line 163: "computed by combining the black and light gray points of the same year" - There are no black points in the plot, I see two sets of grey points. It is unclear why there are two curves and how they are combined to get red points.

- Section 2.2.2: it is unclear how the cloud fraction is included into the forward model.

- Line 179: "For all state vector elements, the OE method requires an a-priori value and its error estimates." - I think "a priori values and their error estimates" would be more correct

- Line 199: "when the measurement errors are assumed to be uncorrelated" - it is unclear if it is the case in this study

- Eq. (2): Can this filter result in skipping real ozone profiles, which are strongly different from a priori, e.g. unexpected ozone loss?

- Line 258: "The full width at half maximum (FWHM) of the AK corresponding to a given altitude is selected in this work as an indicator of the effective vertical resolution of the retrieved profile at this altitude" - Please add here a note, that this approach ignores the displacement of the AK maximum (which you treat then independently as offset).

- Line 258: "the true, physical resolution" - if I understand the meaning of the sentence correctly, the comma needs to be deleted.

- Line 337: "retrieval differences and vertical sampling and smoothing differences" - the wording seems to be incorrect

- Figure 6: A relative difference should be provided in addition.

- Line 359: "cost function > 200" - This criterion was not properly described and justified in Section 2.2

- Figure 9 is unreadable. The panels are too small. There are too many panels in the plot. A rotation by 90 degrees makes the plot even less readable.

- Line 388: "Yearly drifts are added to the temporal dependence plots in Figure 9." - the meaning of this statement is unclear. Are you correcting some data for the drift?

- Figure 10 is not referenced, it is unclear what is the difference between Fig 9 and 10. Figure 10 is unreadable similar to Figure 9.

- Figure 11: The scale of the figure does not enable the reader to estimate the values. The figure cannot definitely be read by people with color vision deficiencies. There are too many panels in one plot. The white dashed line is almost invisible. It cannot be read from the plot if the retrieval compares better than a priori.

- Section 4.4: please state clearly whether the comparisons are done convolving the reference data with the averaging kernels of TROPOMI retrieval.

- Line 431: "... has a mean bias below ±5-10% in the troposphere..." - It is absolutely impossible to see if the mean bias decreases in comparison to the a priori

- Lines 447-450: It looks like this text refers to Figure 10. It is unclear why it is placed after the discussion of Fig. 11.

- Figure A2 needs to be moved to the main text as it is discussed here.

- Line 466: "... on average the observed differences confirm..." - with exception of the stratospheric lidars this is valid only above 20 km. Thus, this statement is not suitable on general.

- Line 466: Again, Figure A2 is discussed in the main text but placed in the Appendix

- Line 481: "All requirements are summarized in Table 2, with the compliance of the operational TROPOMI ozone profile product added" - It is unclear how the values for the table are obtained. From the ozonesonde comparison, for example, it is difficult to understand why authors claim that the accuracy between 12 and 18 km is below 5%.

- Line 486: "This can be seen from Figure 10, with the black lines (average differences) being within the grey areas (SRD requirements)." - I cannot see anything from Figure 10 but this is most probably because of the quality of the figure.

- Line 501: "... typically amounts to about 5%, meaning ..." - From the color scale used in the plot it is difficult to read whether the values in maxima exceed 5%.

---

## Author Comment (AC1)

The authors very much appreciate the reviewers' insightful comments. These have been addressed point-by-point in the author replies and corresponding manuscript updates. Where reviewer comments strongly overlapped, a single author reply has been provided to both reviewers (indicated as such). Note that in between the initial manuscript submission and this updated version, a few additional ozonesonde stations and comparisons have become available to the authors. These have now been included, and tables and figures have been updated accordingly (minor changes, see redline version). This did however not affect the overall conclusions.

1. From Figure 12 (relative differences of total columns between S5P products), I think it is of importance to dig up the sources of differences in the western ocean out of South Africa. It could be either S5P profile measurements miss the biomass-burning signals due to low sensitivity or S5P total ozone retrievals are contaminated. Please give more descriptions.
   Author reply: The comparison provided in the manuscript does not involve the averaging kernel of the products, as the intention was to show qualitatively the consistency between the products. The consistency can also be seen in the timelines in the Figure below, illustrating the total vertical column of the GODFIT (blue line) and of the Ozone Profile product (orange line), in comparison since the beginning of the mission. There is an obvious bias between the two products over the whole period, which we intend to investigate. We are actually preparing another manuscript, which will be soon submitted, focusing specifically on the Ozone Profile algorithm itself.

[Figure]

   Regarding the area pointed out in the South Atlantic Ocean, we believe that this might be due to the different implementation of the climatology in the two products. Both products use the climatology Labow et al. (2015), however for the Ozone Profile a modified version of this climatology is used (as it is also described in the manuscript in section 2.2.3). In order to reduce the stratospheric influence in the troposphere, the original values of the climatology in the troposphere and upper atmosphere are replaced with the median values along the total ozone axis. This specific area is more driven by the troposphere, therefore a-priori differences might cause larger differences, which will be further investigated in the next manuscript on the retrieval algorithm itself.
   Manuscript update: Figure 12 was updated and a phrase in line 526 was added: "A slightly higher bias can be observed in the western ocean out of South Africa, which might be due to the difference in the climatology implementation between

the two products. This is under investigation and will be discussed in a follow-up manuscript."

2. Figure 5: the geophysical distribution of the mean sub-column layers looks very informative for both the lower troposphere and upper stratosphere, showing no apparent artefacts. However, the S5P product is expected to offer high spatiotemporal information compared to other processors (e.g., OMI, GOME). Therefore, I highly encourage you to provide an example of the daily (or a few days average) tropospheric ozone map zoomed in on a specific continent (e.g., Europe, Asia).
Author reply: Only one day of data might contain quite some clouds, therefore we decided to provide a few days average (5 days) over Europe. We applied a cloud filter of 0.2 to look at cloud-free scenes. See manuscript update for description.
Manuscript update: Figure 5 has been updated with the new figure and the caption has been accordingly updated. A description of the new figure has been added to the text: "In addition, Figure 5 (g) shows a five-day average in October 10-15 of the 0-6 km layer. The map data contains a cloud filter to only look at cloud-free scenes (cloud fraction below 0.2). The map clearly shows some regions with higher ozone levels in Eastern Europe and reduced columns over the Alps and the Pyrenees."

3. I would like to recommend moving Section 2.3 to Section 3 as a part of validation meteorology, rather than retrieval meteorology.
Author reply: The authors agree with this suggestion.
Manuscript update: Section 2.3 has been inserted into Section 3 as Section 3.1 (only title marked in red).

4. Line 224: What is the threshold value applied for the quality of fit (cost function) in data screening?
Author reply: 200
Manuscript update: The following sentence is added "It is recommended to apply a screening to the retrieval values showing a cost function fc larger than 200."

5. Line 229-253: the description of the operational validation system is better to be placed in Appendix to ensure the consistency and conciseness in text.
Author reply: The authors tend to disagree with this suggestion. The manuscript appendix is limited to tables and figures, hence having only this text added there would seem not to improve readability. In general, the MPC systems/services are quite complicated to understand their connection and functionalities, so for external people who are interested it can be useful to have a brief description/introduction readily available in this work.
Manuscript update: none

6. This manuscript should be revised overall to enhance readability. Even when the full name of the term is occasionally abbreviated in this paper, there is frequent inconsistency in the use of abbreviations and full names, even when the abbreviation is more familiar to the reader than the full name. DFS, TOLNet, WOUDC, NDACC, AK, FWHM, GAW, SHADOZ.

Author reply: The authors have verified the appearance of all acronyms throughout the manuscript text. The authors however have to stick to the Copernicus journal policy of spelling out acronyms upon first usage.
Manuscript update: Double appearances of full terms and acronyms have been removed at those instances where this was believed to increase readability.

7. (a) Section 3.1 should be placed just before Section 3.2.4 or merged into Section 3.2.4 to ensure the consistency and conciseness in text.
Author reply: The authors tend to disagree with this suggestion. As explained at the beginning of Section 3, the authors are following the validation scheme developed in Keppens et al. (2015). Information content studies are an essential part of this scheme, and therefore situated in the QA approaches, before being discussed independently in Section 4 as well.
Manuscript update: none
(b) And, it is much better to re-organize Section 4 into three parts, Retrieval characteristics including all AK, DFS, FWHM, sensitivity…, Geophysical distribution and validation results (comparison results between S5P and ground reference, between S5P profiles and S5P column).
Author reply: The authors are following the validation scheme developed in Keppens et al. (2015), where this distinction is less strict. It is especially insightful to look at information content measures for the co-located data, as this allows interpreting the observed differences in terms of important retrieval characteristics. In order to nevertheless address the reviewer's request, the authors have added global sensitivity maps to Figure 6 (numbering of initial submission).
Manuscript update: Sensitivity maps are added to Figure 6. Reference to these plots has been added in Section 4.2.
(c) Correspondingly, Figures 9-11 should be re-organized to enhance the readability
Author reply: Both reviewers have expressed major concerns on the readability of Figures 9 to 11, and related Figure A2 in the Appendix. The authors agree with the reviewers' suggestions and have therefore made substantial changes to these Figures and their captions.
Manuscript update: (1) Only the first three columns of Figures 9 and 10 are kept, while the remaining columns are moved to the Appendix. (2) These first three columns are provided in portrait instead of landscape orientation. (3) The SZA colour coding has been removed from Figure 11, although it is maintained in Figure A2 (left column) in the Appendix for the expert reader. (4) A legend is added to Figure 11. (5) The latitude-dependent drift results of Figure A2 (right column) have been moved to a new Figure in the main text.

8. Multi-Figures should have captions (Figure 1.a, 2.b, 3.b).
Author reply: The authors agree and will update the relevant figures.
Manuscript update: The relevant multi-figure plots have been updated.

9. Figure 9 and Figure 11. It is not necessary to perform the retrieval characteristics (DFS, sensitivity, FWHM, offset) at specific stations and specific reference dataset. I think a few orbit files are enough to specify the dependence of the retrieval characteristics on the geophysical parameters (SZA, VZA, cloud fraction…).

Author reply: This comment very much relates with point 7. It is especially insightful to look at information content measures for the co-located data, as this allows interpreting the observed differences in terms of important retrieval characteristics. In order to nevertheless address the reviewer's request, the authors have added meridian sensitivity maps to Figure 6 (numbering of initial submission).
Manuscript update: See point 7.

10. This paper strongly assures that the impact of sampling and smoothing errors on comparison results between S5P and reference are insignificant, thanks to the application of tight co-location criteria and AK smoothing of the reference observations. But, the substantial offset between nominal retrieval altitude and effective vertical retrieval altitude could introduce artificial features, right?
Author reply: The offset between the nominal retrieval altitude and the effective vertical retrieval altitude is also accounted for by averaging kernel smoothing, i.e., upon application of Eq. (3). This has been made clearer in the manuscript update.
Manuscript update: In Section 3.2.4 "Given that the effective vertical resolution of the satellite retrieval is significantly lower than the resolution of the retrieval grid (also see information content studies)..." has been replaced by "Given that the satellite retrievals show an effective vertical resolution and altitude registration that differs from the retrieval grid (also see Sections 3.1 and 4.3)..."

11. (a) Please add legends in Figure 11 for many lines (dashed, dotted, thin). Need to revise Figure 11.
Author reply: The authors agree with this suggestion.
Manuscript update: See update on Figures 9-11.
(b) It is hard to draw any insight on the data assessments as a function of SZA with the current way to put all individual profiles with different color-cording as a function of SZA. Maybe take a look at the mean difference/standard deviations for several SZA regimes (SZA < 40, SZA all, SZA > 60….)
Author reply: This suggestion is already taken into account for the sub-column plots in Figure 9 and 10, and by the overall update of Figures 9-11.
Manuscript update: See update on Figures 9-11.

12. In Section 4.7 and the conclusions (pages 585-595), the authors present a comparative analysis of data quality, comparing their TROPOMI operational ozone profiles with those from other research products (Zhao et al. 2020, Mettig et al. 2021; 2022, Malina et al., 2020), relying on literature assessment reports. Specifically, they assert that, "Apart from these exceptions (Malina et al., 2020 and Mettig et al. products), the operational product demonstrates comparable or lower uncertainty than the scientific products." However, it is not proper to draw definitive conclusions without conducting cross-validation using the same reference and validation criteria. With the difficulty in collecting other S5P ozone profile products for a long-term period, I recommend removing Section 4.7, except for the comparison results between S5P total column and S5P ozone profile-integrated column. Instead, provide a concise summary of other scientific products developed for delivering ozone profile information from TROPOMI measurements in the Introduction.

Author reply: Both reviewers have suggested moving at least part of section 4.7 (especially lines 505–516) to the introduction. The authors agree with this and have updated the manuscript accordingly. On the other hand, the authors prefer keeping the remainder of the section, including the discussion on the validation results of the scientific products. It is acknowledged that validation approaches cannot be fully matched, but this does not usually hamper comparative validation assessments in the literature. In order to take into account the reviewer's concern, the authors have therefore added a disclaimer to this discussion.

Manuscript update: Lines 505-516 have been moved to the introduction, subject to technical corrections. The following disclaimer has been added to Section 4.7 instead: "The operational TROPOMI ozone profile validation results obtained in this work are additionally compared with those of the scientific TROPOMI ozone profile retrieval algorithms that have been found in the literature (see Introduction). However, as the validation approaches for these products are not matched, this comparison should be considered with caution, and within their spatiotemporal validity."

13. Figure 2: what does '" the middle radiance bin" mean? Does it indicate the middle spectral pixel of 270-330 nm, at 290 nm? If so, please delete the middle radiance bin in part of describing Figure for the right panel of Figure 2.

Author reply: In the soft-calibration routine, we compute the correction parameters as a function of several variables, among which also the radiance. The radiance is additionally binned in 20 bins, with each bin showing a particular atmospheric scene (first bins, for example, represent cloudy scenes). In the manuscript, we only give an example of the correction for the central radiance bin. The author agrees that this might cause some confusion as we only wanted to show the general trend of the radiometric correction, therefore the middle radiance bin from the description will be deleted. Detailed information about the correction can be found in the ATBD of the Ozone Profile.

Manuscript update: the description of Figure 2 has been updated accordingly to the answer above, and adding multi figures names ("left" replaced with "(a)"; "right" replaced with "(b)"). From line 162, we also deleted the reference to "the middle radiance bin" and replaced "left" with "(a)", "right" with "(b)"

14. Line 545 "The main elements of the operational retrieval algorithm include the spectral pre-processing, which involves spectral/spatial regridding and wavelength/radiometric correction, a forward model, and an optimal estimation based inverse model."

Author reply: The authors agree that the phrasing of this sentence can be improved, however the reviewer's suggestion is not fully correct as the spectral preprocessing is also one of the pre-processing steps. The authors suggest the following update.

Manuscript update: The main elements of the operational retrieval algorithm include several pre-processing steps, the forward model, and the Optimal Estimation fitting based on the inverse model.

15. Line 553 The reference dataset used here includes WOUDC ozonesondes, TOLN tropospheric lidars, and NDACC stratospheric lidars.

Author reply: The authors agree that a rephrasing of this sentence may improve readability, but the reviewer's suggestion is not fully correct, as ozonesonde data does not only originate from WOUDC.

Manuscript update: "The latter are acquired by ozonesondes contributing to WMO's Global Atmosphere Watch, by tropospheric lidars from the Tropospheric Ozone Lidar Network, and by NDACC stratospheric lidars."

16. Line 508 delete "additionally applied to the MLS and to OMPS for intercomparison", And, connecting to the following sentence, like, "which applied for joint UV-IR retrievals from TROPOMI and CrIS.

    Author reply: The authors agree with this suggestion.

    Manuscript update: The text has been updated as follows: "Mettig et al. (2021) followed with the TOPAS (Tikhonov regularised Ozone Profile retrievAl with SCIATRAN) algorithm, which has also been used for joint UV-IR retrievals from TROPOMI and the Cross-track Infrared Sounder on the Suomi National Polar-orbiting Partnership (CrIS/Suomi-NPP)." Note that these lines (505-516) have been moved to the introduction.

17. Line 515 delete ",which is applied ~ data"

    Author reply: The authors agree with this suggestion.

    Manuscript update: This sentence part has been removed from the text. Note that these lines (505-516) have been moved to the introduction.

18. C17. Line 547 ranging from ➜ spanning

    Author reply: We agree with this suggestion.

    Manuscript update: "spanning"

19. Line 580. Please specify "Observed above" e.g. observed for the stratospheric ozone retrievals.

    Author reply: We agree with this suggestion.

    Manuscript update: "above" is replaced by "for the stratospheric ozone retrievals"

20. Line 584 Please provide any reference to "this agrees with the operational TROPOMI total ozone column retrieval". And, the long-term stability is commonly assured for other S5P L2 products?

    Author reply: The authors agree with the suggestion to add a reference. On the other hand, it is hard to compare the TROPOMI (operational) ozone profile drift with the drift of other molecule retrievals. Only the O3_PR retrieval makes use of TROPOMI's UV spectrometer, which degrades differently (typically faster) than the other detectors within the instrument.

    Manuscript update: After this statement, a reference to Garane et al. (2019) has been added to the text.

---

## Author Comment (AC2)

The authors very much appreciate the reviewers' insightful comments. These have been addressed point-by-point in the author replies and corresponding manuscript updates. Where reviewer comments strongly overlapped, a single author reply has been provided to both reviewers (indicated as such). Note that in between the initial manuscript submission and this updated version, a few additional ozonesonde stations and comparisons have become available to the authors. These have now been included, and tables and figures have been updated accordingly (minor changes, see redline version). This did however not affect the overall conclusions.

Unfortunately, in Section 4, which describes the retrieval sensitivity and validation results, the authors decided to save space and joined tens of panels with thousands of curves into single plots. This makes the plots absolutely unreadable (Figs 9, 10, 11 and A2). Although the text is still well written, the reader cannot follow the discussion and verify the conclusions of the authors because no information can be read from the plots. In my opinion, the authors have to reconsider the way they present their results in Section 4 to make the manuscript suitable for publication. It should be analyzed which information is important and how to present it without making the plots look like squeezed together unresolved color spots. A rotation of the plots by 90 degrees is also not a really good idea. This makes the reader terribly difficult to look at the plots when reading the text.

Author reply: Both reviewers have expressed major concerns on the readability of Figures 9 to 11, and related Figure A2 in the Appendix. The authors agree with the reviewers' suggestions and have therefore made substantial changes to these Figures and their captions.

Manuscript update: (1) Only the first three columns of Figures 9 and 10 are kept, while the remaining columns are moved to the Appendix. (2) These first three columns are provided in portrait instead of landscape orientation. (3) The SZA colour coding has been removed from Figure 11, although it is maintained in Figure A2 (left column) in the Appendix for the expert reader. (4) A legend is added to Figure 11. (5) The latitude-dependent drift results of Figure A2 (right column) have been moved to a new Figure in the main text.

- The paragraph mentioning previous retrievals (lines 505–516, Section 4.7) should be moved to the introduction.
  Author reply: Both reviewers have suggested moving at least part of section 4.7 (especially lines 505–516) to the introduction. The authors agree with this and have updated the manuscript accordingly.
  Manuscript update: Lines 505-516 have been moved to the introduction, subject to technical corrections.
- Figure 1: the right box shows an endless loop between the radiative transfer and optimal estimation
  Author reply: It is not an endless loop in practice. Inside the box is written that it is until the convergence of eight iterations at maximum.
  Manuscript update: The following is added to the caption: "until convergence or the maximum number of eight iterations is reached."
- Line 125: for the pixel size after August 6, 2019 please indicate which size is crosstrack and which along-track.
  Author reply: (across-track x along-track)
  Manuscript update: This has been implemented in the text.

- Line 147: Usage of the CAMS data, with assimilated MLS profiles, and scaling them to OMPS total columns is expected to give a very good approximation for the ozone profile. In this respect the author should put a bit more focus to show that TROPOMI measurements increase the information content in comparison to a priori. Maybe it is already shown in plots in Section 4, they are, however, completely unreadable for me.

  Author reply: This is not the a-priori profile, which is described in section 2.2.3 and which comes from a modified version of the climatology references (Labow et al.). We use this ozone profile in the soft-calibration routine to compute the radiance expected from the best ozone profile shape we could get from that specific location. Then, we compare this estimate with a forward model calculation to compute the soft-calibration parameters.

  Manuscript update: This sentence was added: "Figure 6 clearly shows how the a-priori is smoothed by the measurements in the retrieved ozone. Moreover, the vertical sensitivity shown at the bottom indicates that the measurements add more information with respect to the a-priori in most of the vertical layers, with low-sensitivity areas depending on the latitude (as also shown in Figure 11)."

- Line 163: "computed by combining the black and light gray points of the same year" - There are no black points in the plot, I see two sets of grey points. It is unclear why there are two curves and how they are combined to get red points.

  Author reply: Yes, the image was not the one described and it has been updated with the correct one showing the points with different colours.

  Manuscript update: Now the points show the two different colours which refer to the two different datasets (OFFL and RPRO) combined to compute the total correction. This is also indicated in the plot legend.

- Section 2.2.2: it is unclear how the cloud fraction is included into the forward model.

  Author reply: The authors agree that a manuscript update is needed.

  Manuscript update: The following sentence is added to the manuscript: "In the forward model, clouds are represented as Lambertian reflecting surfaces which cover part of the ground pixel and are placed at cloud pressure. The cloud pressure and fraction are derived from the FRESCO algorithm using the oxygen A-band of TROPOMI at 760 nm. The cloud fraction and albedo are fitted at 330 nm during the retrieval of the ozone profile."

- Line 179: "For all state vector elements, the OE method requires an a-priori value and its error estimates." - I think "a priori values and their error estimates" would be more correct

  Author reply: The authors agree with this suggestion.

  Manuscript update: The sentence has been updated in agreement.

- Line 199: "when the measurement errors are assumed to be uncorrelated" - it is unclear if it is the case in this study.

  Author reply: The error covariance matrix of the measurements is diagonal if there are no calibration errors. The diagonal then contains only the errors dominated by shot noise and are uncorrelated. In the current version of the retrieval algorithm, the non-diagonal part of the matrix is set to zero, therefore no calibration errors are considered.

  Manuscript update: "when" is replaced by "here as"; plus the phrase "dominated by shot noise and therefore" is added to specify our case.

- Eq. (2): Can this filter result in skipping real ozone profiles, which are strongly different from a priori, e.g. unexpected ozone loss?

Author reply: This filter has been applied to avoid unphysical deviation of the retrieval from the a-priori. It is not excluded that it can contain false positives, as any data screening would have. However, it can be seen in Figure 6 (in the initial submission) that the absolute difference between the ozone profile retrieval and a-priori is typically an order of magnitude smaller than this filter value, which would be equal to 10 on the current plot scale. It can also be seen from the third column in Appendix A1, that this filter is the one removing the least pixels.
Manuscript update: none

- Line 258: "The full width at half maximum (FWHM) of the AK corresponding to a given altitude is selected in this work as an indicator of the effective vertical resolution of the retrieved profile at this altitude" - Please add here a note, that this approach ignores the displacement of the AK maximum (which you treat then independently as offset).
Author reply: We agree with this suggestion.
Manuscript update: "The full width at half maximum (FWHM) of the AK corresponding to a given altitude is selected in this work as an indicator of the effective vertical resolution of the retrieved profile at this altitude, although it is determined independently of any vertical displacement of the kernel"

- Line 258: "the true, physical resolution" - if I understand the meaning of the sentence correctly, the comma needs to be deleted.
Author reply: We agree with this suggestion.
Manuscript update: Comma is deleted.

- Line 337: "retrieval differences and vertical sampling and smoothing differences" - the wording seems to be incorrect
Author reply: We agree that this phrasing can be improved.
Manuscript update: "retrieval differences including vertical sampling and smoothing differences"

- Figure 6: A relative difference should be provided in addition
Author reply: The authors agree with this suggestion, but the absolute difference has been removed from the plot to avoid having a large figure. Only the relative difference is provided now.
Manuscript update: The absolute difference has been replaced by the relative difference.

- Line 359: "cost function > 200" - This criterion was not properly described and justified in Section 2.2.
Author reply: The manuscript has been updated in the section regarding data selection.
Manuscript update: The following sentence is added "It is recommended to apply a screening to the retrieval values showing a cost function fc larger than 200."

- Figure 9 is unreadable. The panels are too small. There are too many panels in the plot. A rotation by 90 degrees makes the plot even less readable.
Author reply: The authors agree with this suggestion.
Manuscript update: See update on Figures 9-11.

- Line 388: "Yearly drifts are added to the temporal dependence plots in Figure 9." - the meaning of this statement is unclear. Are you correcting some data for the drift?
Author reply: The authors agree that this statement is dubious. The text has therefore been rephrased and extended.

Manuscript update: "Yearly drift values that are calculated from a linear fit are added to the temporal dependence plots in Figure 9. The two-sigma uncertainties on these values result from a bootstrapping technique with thousand samples (Efron and Tibshirani, 1986)."

- Figure 10 is not referenced, it is unclear what is the difference between Fig 9 and 10. Figure 10 is unreadable similar to Figure 9.
  Author reply: Because of their similarity, Figures 9 and 10 are kept together, although Figure 11 appears in the text before Figure 10.
  Manuscript update: See update on Figures 9-11. The discussion of these figures in the text has been updated accordingly.
- Figure 11: The scale of the figure does not enable the reader to estimate the values. The figure cannot definitely be read by people with color vision deficiencies. There are too many panels in one plot. The white dashed line is almost invisible. It cannot be read from the plot if the retrieval compares better than a priori.
  Author reply: The authors agree with this suggestion.
  Manuscript update: See update on Figures 9-11. It has additionally been stressed how the retrieval performs in comparison to the mean prior in the first paragraph of Section 4.4.
- Section 4.4: please state clearly whether the comparisons are done convolving the reference data with the averaging kernels of TROPOMI retrieval.
  Author reply: The authors agree with this suggestion.
  Manuscript update: The first sentence of Section 4.4 has been updated as follows (initial numbering): "Figure 11 contains all comparisons between TROPOMI ozone profiles and reference data, the latter AK-smoothed using Eq. (3), and corresponding statistics."
- Line 431: "... has a mean bias below ±5-10% in the troposphere..." - It is absolutely impossible to see if the mean bias decreases in comparison to the a priori
  Author reply: The authors agree with this suggestion.
  Manuscript update: See update on Figures 9-11. It has additionally been stressed how the retrieval performs in comparison to the mean prior in the first paragraph of Section 4.4.
- Lines 447-450: It looks like this text refers to Figure 10. It is unclear why it is placed after the discussion of Fig. 11.
  Author reply: Because of their similarity, Figures 9 and 10 are kept together, although Figure 11 appears in the text before Figure 10.
  Manuscript update: See update on Figures 9-11. The discussion of these figures in the text has been updated accordingly.
- Figure A2 needs to be moved to the main text as it is discussed here
  Author reply: The authors agree with this suggestion.
  Manuscript update: See update on Figures 9-11.
- Line 466: "... on average the observed differences confirm..." - with exception of the stratospheric lidars this is valid only above 20 km. Thus, this statement is not suitable in general.
  Author reply: The authors agree that the "on average" at the beginning of the sentence may be confusing. It is now stressed that this only applies to the stratosphere.
  Manuscript update: Line 466 (initial numbering) has been updated as follows: "The chi-square plots in Figure 11 (third graph in each plot) demonstrate that

the observed differences confirm (chi^2 close to one) the combined ex-ante satellite and ground uncertainty estimates in the stratosphere on average, despite the appearance of large outliers."

- Line 466: Again, Figure A2 is discussed in the main text but placed in the Appendix
  Author reply: The authors agree with this suggestion.
  Manuscript update: See update on Figures 9-11.
- Line 481: "All requirements are summarized in Table 2, with the compliance of the operational TROPOMI ozone profile product added" - It is unclear how the values for the table are obtained. From the ozonesonde comparison, for example, it is difficult to understand why authors claim that the accuracy between 12 and 18 km is below 5 %.
  Author reply: The authors agree that this is hard to see from Figure 10 (in its initial formatting), which contains grey areas indicating the product requirements on the uncertainty. These should be more clearly visible by the update of Figures 9-11.
  Manuscript update: See update on Figures 9-11.
- Line 486: "This can be seen from Figure 10, with the black lines (average differences) being within the grey areas (SRD requirements)." - I cannot see anything from Figure 10 but this is most probably because of the quality of the figure.
  Author reply: The authors agree that this is hard to see from Figure 10 (in its initial formatting), which contains grey areas indicating the product requirements on the uncertainty. These should be more clearly visible by the update of Figures 9-11.
  Manuscript update: See update on Figures 9-11.
- Line 501: "... typically amounts to about 5%, meaning ..." - From the color scale used in the plot it is difficult to read whether the values in maxima exceed 5%.
  Author reply: The authors agree with this comment.
  Manuscript update: Figure 12 has been updated with extended colour scale.

---

## Referee Report (RR1)

**Referee report to the revised version of the "Five years of Sentinel-5p TROPOMI operational ozone profiling and geophysical validation using ozonesonde and lidar ground-based networks" manuscript by Arno Keppens et al.**

The manuscript has been significantly improved with respect to the presentation quality. Most of my comments were addressed in a satisfactory way. However, some issues are still needed to be dealt with. The manuscript can be accepted for the publication in AMT after a minor revision. My detailed comments are provided below.

**Detailed comments**

- Lines 13-14: "vertical sensitivity" - it is not a common notation and should be defined before using.

- Line 20: "meridian dependence of its bias" - Whose bias is meant here, that of the sensitivity or of the tropospheric ozone?

- Line 66: "The same combination of TROPOMI UV and CrIS IR retrieval wavelengths has been exploited by ..." - "A similar" instead of "the same" would be more correct, as the wavelength ranges used by these two retrievals are quite different.

- Line 160: "Additionally, the CAMS ozone profiles are scaled to match the total ozone column derived from the OMPS total column data (Jaross, 2017)." - I am wondering why you do not use the total ozone column from TROPOMI instead (just for a curiosity, not as a requirements to change).

- Line 267: Could you please comment on the value of 200 for the cost function threshold. How did you come to this value?

- Lines 341-342: "... resolution and altitude registration that differs from the retrieval grid ..." - what does "altitude registration" mean here? You probably want to highlight what the AK peaks are not at nominal altitudes but this formulation seems quite confusing to me.

- Line 382: "... the a-priori is smoothed by the measurements ...." - this statement sounds extremely confusing. I am sure you agree, measurements cannot affect a priori in any way. Please reword.

- Line 473: "... an increase of the DFS ..." - Fig. A3 does not show any DFS, I suppose you refer to Fig. A2 here. From the sentence it is not clear if you refer to 6-12 km column then talking about DFS increase with SZA. Looking at Fig. A2 I see a much larger increase of DFS with SZA for the 12-18 km column (the third row from the bottom) than for the 6 - 12 km one.

- Line 473: "... an increase of the ... bias for the 6-12 km column with SZA" - In Fig. 3 I do not see any increase of the bias with SZA for any of the columns.

- Lines 473 - 475: "This correlation seems to be somewhat compensated for in the lowest column by increased atmospheric penetration of the sunlight at low solar zenith angles (0 to about 30°)." - I cannot understand which correlation you are talking about here and where you see it compensated.

- Lines 475 - 477: "Additionally, the bias is clearly negatively correlated with the surface albedo for the 6-12 km subcolumn, despite the latter's apparently slightly positive correlation with the retrieval DFS." - a similar correlation for the differences is seen for the 0-6 km column and a bit reduced for the 12 - 18 km column. DFS for 0 - 6 km column does not seem to show any correlation with albedo while this correlation for the 12-18 km is largest. In general this sentence does not seem to overview the full picture.

- Lines 481 - 482: "... while a negative drift is observed for the two subcolumns above (18-32 km)." - I see a negative drift only for 18 - 24 km but not for 24-32 km (numbers in the pot), are you still discussing Fig 10? By the way, in the caption of Fig. 10 it is not explicitly indicated which column belongs to which row. I understand it is the same as for Fig 9 but this still should be mentioned explicitely.

- Lines 519 - 520: "This can be seen from Figure 10, with the black lines (average differences) being within the grey areas (SRD requirements)." - This is not really visible in the plot, especially in the right column.

- Line 527: "The vertical retrieval grid is sampled at a resolution of 6 km or higher, ...." and Table 2: "Partially, as the vertical grid complies, ..." - I do not think it is correct to rate a sufficient sampling of the vertical grid as a partial compliance with respect to the vertical resolution. I agree it is required to have a vertical grid with a sufficient sampling but it has nothing to do with the measurement/retrieval capabilities.

- Line 536: "... observed in the western ocean out of South Africa ..." - Do you mean "in the Atlantic ocean western of South Africa"?

- Figure 9: suboptimal position of the text boxes in the lower right plots, the boxes strongly cover the plot contents. 50% quantile lines are often difficult to distinguish, another color, e.g. green, might help.

- Figure 11: The figure is still difficult to read. It should be stretched to occupy the full page width. Horizontal space between the sub-plots would be useful

- Figure A4: same as for Fig. 11

**Technical corrections:**

- Line 19: "The vertical sensitivity of the TROPOMI tropospheric ozone amount" - This sounds a bit weird to me. Maybe you should exchange "of" by "to" or "for", or talk about sensitivity of the retrieval and not that of ozone amount...

- Line 268: "...for all 33 levels $l$ combined..." - should "$l$" be separated by commas?

- Line 302: "...of up to 5 %, and except in the tropical upper troposphere,..." - should there be a comma after "and"?

---

## Author Response (AR2)

Editor comments:

l. 185: It should be clarified if the retrieval uses forward calculations with and without clouds, using the cloud fraction as weights for partially cloudy scenes.

Author reply: Yes, this is the way it is computed internally in the radiative transfer code (DISAMAR software) used for the forward model calculations.

Manuscript update: The following phrase has been added: "The computations are performed with and without clouds, using the cloud fraction as a weight for partially clouded scenes." And other minor changes in the paragraph to improve the readability.

l. 198: "along the total ozone axis" sounds awkward, maybe say "in both ranges the median from all profiles for a given total ozone value is used.

Author reply: We agree that the expression might not be clear; therefore we have accordingly updated the manuscript.

Manuscript update: On line 195: we have replaced the phrase "with the median of all the values along the total ozone axis" with "with the median ozone profiles for a given total ozone value."

l. 369: do not use the term "sub-columns". Mention panel numbers instead. You may say "the panels on the left and right sides".

Author reply: We agree that the phrasing on sub-columns (here of ozone, not of the Figure) can be confusing.

Manuscript update: On line 363 (in the numbering of the new version), "sub-column" has been replaced by "panel".

Fig. 2: Replace orbit numbers with a time axis, e.g. years (or add a second time axis). Technically, each individual plot needs a separate panel number.

Author reply: The authors agree with the suggestion.

Manuscript update: A secondary time axis has been added on the left panel, while in the right panel the orbits in the legend have been replaced with the years of the mission. The figure caption has also been modified in accordance to the new numbering of each individual plot.

Fig. 5: use larger fonts in the maps

Manuscript update: A new figure has been updated to the manuscript, with increased fonts and each panel having a label.

Fig. 6: each plot needs a panel number

Manuscript update: The figure is updated, with each panel having a label.

Fig. 11: use a lighter grey colour in the plot (too dark). Add panel numbers for each plot (three) and mention the ground-based measurement type in each panel. Explain what X (chi) is.

Author reply: These suggestions have been taken into account upon recreating the plots. The mathematical formulation for chi square - as used in the plot - has been added to the figure caption for clarity, but it has not been further explained, as this would then be required for all plotted quantities.

Manuscript update: Figures 11 and A4 have been horizontally stretched to increase readability, and horizontal scales of absolute and relative differences have been updated accordingly (larger range hence showing more white background). The reference instrument type has been added to the title text. A lighter shade of grey has been used for the individual line plots (only applicable to Figure 4). Panel numbers have been added for referencing in the respective captions. The mathematical formulation for chi square - as used in the plot - has been added to the figure caption for clarity.

Fig. 14: Make the fonts larger in each map. In the figure caption, explain the panels in the right order (now: you start with c and then a, b). Alternatively, you can reorder the maps to match the order in the figure caption.

Author reply: The suggestion is taken into account and the text and the figure have been accordingly updated (Fig. 13 in the new file).

Manuscript update: The figure caption has been replaced to match the order of the panels.

Fig. A4: see comments to Fig. 11.

Author reply: see reply to comments on Fig. 11.

The appendix should be separated from the paper and be available as a supplement. Then, the numbering should be S1 instead of A1, and so on.

Author reply: This had actually been our first intention, but upon checking the AMT Supplements guidelines, this did not seem to be the best approach (https://www.atmospheric-measurement-techniques.net/submission.html): "Supplementary material is reserved for items that cannot reasonably be included in the main text or as appendices. These may include short videos, very large images, maps, CIF files, as well as short computer codes such as Matlab or Python script." Especially for the reference station lists, an Appendix with direct access to detailed information seems more appropriate to expert readers. We will however comply with the eventual publication instructions of the Copernicus / Latex editor.

Manuscript update: TBD with the Copernicus editor upon manuscript acceptance.

Referee comments:

Referee report to the revised version of the "Five years of Sentinel-5p TROPOMI operational ozone profiling and geophysical validation using ozonesonde and lidar ground-based networks" manuscript by Arno Keppens et al. The manuscript has been significantly improved with respect to the presentation quality. Most of my comments were addressed in a satisfactory way. However, some issues are still needed to be dealt with. The manuscript can be accepted for the publication in AMT after a minor revision. My detailed comments are provided below.

Detailed comments:

• Lines 13-14: "vertical sensitivity" - it is not a common notation and should be defined before using.

Author reply: This term is explained between brackets in the manuscript update.

Manuscript update: "(i.e., the fraction of the information that originates from the measurement)" has been added right after "vertical sensitivity".

• Line 20: "meridian dependence of its bias" - Whose bias is meant here, that of the sensitivity or of the tropospheric ozone?

Author reply: The authors agree that this phrasing is confusing.

Manuscript update: "its" has been replaced by "the".

• Line 66: "The same combination of TROPOMI UV and CrIS IR retrieval wavelengths has been exploited by ..." - "A similar" instead of "the same" would be more correct, as the wavelength ranges used by these two retrievals are quite different.

Author reply: The authors agree that the rephrasing is more correct.

Manuscript update: The update has been implemented as suggested.

• Line 160: "Additionally, the CAMS ozone profiles are scaled to match the total ozone column derived from the OMPS total column data (Jaross, 2017)." - I am wondering why you do not use the total ozone column from TROPOMI instead (just for curiosity, not as a requirement to change).

Author reply: This is for historic reasons. In the beginning of the mission, we did not have a well validated total ozone L3 product available, and the procedure has not been changed to allow for consistency. In the future, when we reprocess the soft calibration data, this could be updated to use TROPOMI total column data.

Manuscript update: None

• Line 267: Could you please comment on the value of 200 for the cost function threshold. How did you come to this value?

Author reply: This value has been determined from sensitivity studies in order to have an indication for the users on how to remove data with poor fitting quality. As explained in the PRF (Product

Readme File) of the Ozone Profile product, this value is used as a diagnostic information for the ground pixels showing reduced quality of the fit.

Manuscript update: on line 264, "(value determined from sensitivity studies)".

• Lines 341-342: "... resolution and altitude registration that differs from the retrieval grid ..." - what does "altitude registration" mean here? You probably want to highlight that the AK peaks are not at nominal altitudes but this formulation seems quite confusing to me.

Author reply: The authors agree that this phrasing may sound unnecessarily complex. It has now been simplified in agreement with Section 4.3.

Manuscript update: "an effective vertical resolution and altitude registration" has been replaced by "an effective vertical position and resolution" for "each retrieved ozone value".

• Line 382: "... the a-priori is smoothed by the measurements ...." - this statement sounds extremely confusing. I am sure you agree, measurements cannot affect a priori in any way. Please reword.

Author reply: The authors agree with the rephrasing suggestion. It was not meant that the measurements affect the a-priori but more how the measurement combines with the a-priori values to give the result visible in the retrieval panel, which does not show the same steps visible in the ozone layer in the a-priori panel.

Manuscript update: The phrase has been replaced with "how the measurements combine with the a-priori values in a smoother retrieved ozone layer at the top panel." The caption of the figure has also been modified according to the new numbering.

• Line 473: "... an increase of the DFS ..." - Fig. A3 does not show any DFS, I suppose you refer to Fig. A2 here. From the sentence, it is not clear if you refer to the 6-12 km column when talking about DFS increase with SZA. Looking at Fig. A2 I see a much larger increase of DFS with SZA for the 12-18 km column (the third row from the bottom) than for the 6 - 12 km one.

• Line 473: "... an increase of the ... bias for the 6-12 km column with SZA" - In Fig. A3 I do not see any increase of the bias with SZA for any of the columns.

• Lines 473 - 475: "This correlation seems to be somewhat compensated for in the lowest column by increased atmospheric penetration of the sunlight at low solar zenith angles (0 to about 30°)." - I cannot understand which correlation you are talking about here and where you see it compensated.

• Lines 475 - 477: "Additionally, the bias is clearly negatively correlated with the surface albedo for the 6-12 km subcolumn, despite the latter's apparently slightly positive correlation with the retrieval DFS." - a similar correlation for the differences is seen for the 0-6 km column and a bit reduced for the 12 - 18 km column. DFS for the 0-6 km column does not seem to show any correlation with albedo while this correlation for the 12-18 km is largest. In general, this sentence does not seem to overview the full picture.

Author reply: These comments are all on the same paragraph, and have therefore been addressed combinedly. All subcolumns of relevance for the ozonesonde comparisons (i.e., below 32 km) were

confusingly indicated as "lowest subcolumns". This has now been corrected for, and the abstract and conclusions have been minimally updated accordingly.

Manuscript update: This paragraph has been rewritten as follows: "An optical path length dependence of the TROPOMI bias is observed for the subcolumns, which also translates into a seasonal and meridian dependence of the bias, as seen in Figure 10. Scatter plots in the Appendix show the dependence of the subcolumn DFS (Figure A2) and bias (Figure A3) on SZA, VZA (both related to path length), cloud fraction, and surface albedo. The bias is clearly negatively correlated with the surface albedo for the lowest three subcolumns, despite the albedo's apparently slightly positive correlation with the retrieval DFS. The meridian dependence of the full profile bias with respect to ozonesondes is shown in Figure A4 for five latitude bands, where increased tropospheric biases are observed for high solar zenith angles in the mid to high latitudes. As a result, increased tropospheric biases are found for high-SZA observations above highly reflective scenes, like is the case for Antarctic (sea) ice. On the other hand, when the deviation from the prior profile becomes too strong, these observations are flagged by the check in Eq. (2)."

• Lines 481 - 482: "... while a negative drift is observed for the two sub-columns above (18-32 km)." - I see a negative drift only for 18 - 24 km but not for 24-32 km (numbers in the plot), are you still discussing Fig 10?

Author reply: Thanks for noticing this error. The 18-24 km column drift was just negative in the first version, and just positive with the ozonesonde data updates for the second version of the plots, while the text had not been updated accordingly. As the drift for this column is negligible, the text now focuses on the significant drifts above and below.

Manuscript update: The quoted text has been corrected as follows: "…and a negative drift of similar size for the 24-32 km subcolumn, while the drift is negligible for the subcolumn in between (18-24 km)."

• By the way, in the caption of Fig. 10 it is not explicitly indicated which column belongs to which row. I understand it is the same as for Fig 9 but this still should be mentioned explicitly.

Author reply: The column ranges that are in the caption of Figure 9 were not repeated for Figure 10 because the latter's y-labels contain this information.

Manuscript update: For completeness, the column ranges have now been added in the caption of Figure 10 as well.

• Lines 519 - 520: "This can be seen from Figure 10, with the black lines (average differences) being within the grey areas (SRD requirements)." - This is not really visible in the plot, especially in the right column.

Author reply: The black lines are not fully visible in the subplots on the right, but are constant within each row and hence can be interpreted from the middle plots. This has been specified in the updated text.

Manuscript update: "black lines (average differences)" is updated to "thick horizontal black lines (average differences that are constant for each row)".

• Line 527: "The vertical retrieval grid is sampled at a resolution of 6 km or higher..." and Table 2: "Partially, as the vertical grid complies..." - I do not think it is correct to rate a sufficient sampling of the vertical grid as a partial compliance with respect to the vertical resolution. I agree it is required to have a vertical grid with sufficient sampling but it has nothing to do with the measurement/retrieval capabilities.

Author reply: The authors agree with the reviewer's concern. However, as the requirements do not differentiate between 'sampling' resolution' and 'effective' resolution, we have attempted to appropriately address both at the same time. As such, we have tried to find a middle ground between an expert versus non-expert user perspective on this matter.

Manuscript update: None.

• Line 536: "... observed in the western ocean out of South Africa ..." - Do you mean "in the Atlantic ocean western of South Africa"?

Author reply: Yes.

Manuscript update: Rephrase has been implemented.

• Figure 9: suboptimal position of the text boxes in the lower right plots: the boxes strongly cover the plot contents. 50 % quantile lines are often difficult to distinguish, another color, e.g. green, might help.

Author reply: The authors agree on the suboptimal position of the legends. Where appropriate, these have been moved to the upper right corner of the panel. In order not to introduce yet another colour in this figure, a different plotting thickness has been used to clearly distinguish between the median and other quantile lines (also for Figures 10, A2, and A3).

Manuscript update: The trend line legends in the lowest three panel rows have been moved to the upper right corner of the panel. A different plotting thickness has been used for the median versus other quantile lines. The figure caption has been updated accordingly (also for Figures 10, A2, and A3).

• Figure 11: The figure is still difficult to read. It should be stretched to occupy the full page width. Horizontal space between the sub-plots would be useful.

• Figure A4: same as for Fig. 11.

Author reply: The reviewer suggestions have been taken into account upon recreating the plots.

Manuscript update: Figures 11 and A4 have been horizontally stretched to increase readability, and horizontal scales of absolute and relative differences have been updated accordingly (larger range hence showing more white background).

Technical corrections:

• Line 19: "The vertical sensitivity of the TROPOMI tropospheric ozone amount" - This sounds a bit weird to me. Maybe you should exchange "of" by "to" or "for", or talk about sensitivity of the retrieval and not that of ozone amount…

Author reply: The authors agree that this phrasing is confusing.

Manuscript update: "amount" has been replaced by "retrieval".

• Line 268: "...for all 33 levels l combined…" - should "l " be separated by commas?

Author reply: This seems to depend on the stress you want to give. For clarity, "l" has been put between brackets, in agreement with the number density symbols in the same line.

Manuscript update: "(denoted l)" has been added to the text for clarity.

• Line 302: "...of up to 5 %, and except in the tropical upper troposphere..." – should there be a comma after "and"?

Author reply: Adding a comma would indeed make the phrasing more correct, but we have opted to split the sentence for readability instead.

Manuscript update: "…of up to 5 %. Except for the tropical upper troposphere…"